# Rethinking Reward Models for Multi-Domain Test-Time Scaling

**Dong Bok Lee[1]***, **Seanie Lee[1]***, **Sangwoo Park[1]**, **Minki Kang[1]**, **Jinheon Baek[1]**,
**Dongki Kim[1]**, **Dominik Wagner[3]**, **Jiongdao Jin[1]**, **Heejun Lee[1,4]**, **Tobias Bocklet[3]**,
**Jinyu Wang[2]**, **Jingjing Fu[2]**, **Sung Ju Hwang[1,4]**, **Jiang Bian[2]**, **Lei Song[2]**
*[1]KAIST   [2]Microsoft Research Asia   [3]TH Nürnberg   [4]DeepAuto.ai*
*markhi@kaist.ac.kr*

**Reviewed on OpenReview:** *https://openreview.net/forum?id=PgouBhL7IR*

## Abstract

The reliability of large language models (LLMs) during test-time scaling is often assessed with *external verifiers* or *reward models* that distinguish correct reasoning from flawed logic. Prior work has studied both outcome reward models (ORMs), which assess only the final answer, and process reward models (PRMs), which score intermediate reasoning steps. Although PRMs are often viewed as advantageous due to their finer-grained supervision, much of the supporting evidence comes from math-adjacent settings, and their relative benefits across broader domains remain unclear. We present the first unified evaluation of four reward model variants, discriminative ORM and PRM (dORM, dPRM) and generative ORM and PRM (gORM, gPRM), across 14 diverse domains. Contrary to conventional wisdom, we find that (i) dORM performs on par with dPRM, (ii) gPRM is not competitive, and (iii) overall, gORM is the most robust, yielding significant and consistent gains across every tested domain. We attribute this to PRM-style stepwise scoring, which inherits label noise from LLM auto-labeling and has difficulty evaluating long reasoning trajectories, including those involving self-correcting reasoning. Our theoretical analysis shows that step-wise aggregation compounds errors as reasoning length grows, and our empirical observations confirm this effect. These findings challenge the prevailing assumption that fine-grained supervision is always better and support generative outcome verification for multi-domain deployment. We publicly release our code at this Github repository to facilitate future research in multi-domain settings.

## 1 Introduction

Test-time scaling (TTS) enables large language models (LLMs) to generate diverse, reliable solutions via chain-of-thought reasoning (CoT; Wei et al., 2022; Kojima et al., 2022; Yao et al., 2023b; Madaan et al., 2023) and has shown impressive results on challenging reasoning tasks (Yao et al., 2023a; Snell et al., 2025; Wu et al., 2024). A widely adopted TTS approach uses *external verifiers* that select the best among the candidates (Snell et al., 2025). One common instantiation, outcome reward models (ORMs), is typically implemented as discriminative classifiers that assign a scalar *reward* to a CoT (Cobbe et al., 2021; Uesato et al., 2022; Yu et al., 2024). ORMs are trained only on outcome-level signals, which are often coarse. Recent work has introduced process reward models (PRMs; Lightman et al., 2024; Wang et al., 2024a; Setlur et al., 2025; Zheng et al., 2024) that score each step of a CoT and aggregate the scores into a trajectory-level reward. Supervised with high-quality, carefully constructed process labels, *e.g.*, manual annotation (Lightman et al., 2024) or Monte Carlo rollouts (Wang et al., 2024a), PRMs have often been shown to outperform ORMs when combined with TTS.

---

*Equal Contribution.

Beyond discriminative verifiers, several studies have shown that the generative ability of LLMs can improve CoT verification, such as *LLM-as-a-judge* (Wang et al., 2023; Liu et al., 2023; Zheng et al., 2023). Based on this idea, other works fine-tune LLMs to generate a verification rationale for a CoT and compute the final reward from token probabilities (Zhang et al., 2025a; Khalifa et al., 2025; Zhao et al., 2025). To obtain verification CoTs for training, most previous work adopts *consensus-filtering*: (i) generate verification CoTs, and (ii) retain the verification CoT if its parsed verdict aligns with outcome or process labels. After training, these generative verifiers have shown strong performance in math-adjacent reasoning tasks, outperforming discriminative verifiers.

However, most of the research efforts on TTS with external verifiers have been devoted primarily to math-adjacent domains. This narrow scope limits the potential for LLM deployment in high-stakes real-world applications, such as the legal (Guha et al., 2023; Cui et al., 2023; Fei et al., 2024) and medical (Singhal et al., 2023; Kung et al., 2023; Singhal et al., 2025) domains, where trustworthiness is paramount and rigorous verification of LLM outputs is especially important. Recently, Zeng et al. (2025) proposed multi-domain PRMs trained on the graduate level benchmark (MMLU-Pro; Wang et al., 2024c), covering 14 diverse domains, and showed that multi-domain training for PRMs significantly improves TTS performance across diverse domains. However, the study is *limited* to discriminative PRMs and the broader potential of different verifier types (*e.g.*, ORMs vs. PRMs, discriminative vs. generative) in the multi-domain setting remains *underexplored.*

To this end, we present the first controlled multi-domain TTS evaluation of four verifier variants, discriminative ORM and PRM (dORM, dPRM), and generative ORM and PRM (gORM, gPRM), trained and evaluated under a common protocol across 14 diverse domains. We review these variants in §2 and, under controlled conditions, evaluate them on math (PRM800K, ProcessBench; Lightman et al., 2024; Zheng et al., 2024), multi-domain (MMLU-Pro; Wang et al., 2024c), and specialized-domain (GPQA-Diamond, MedQA, LEXam; Rein et al., 2024; Jin et al., 2021; Fan et al., 2026) in §3. In the math domain, trends across the four variants are consistent with prior work (Lightman et al., 2024; Zhang et al., 2025a; Khalifa et al., 2025). dPRM

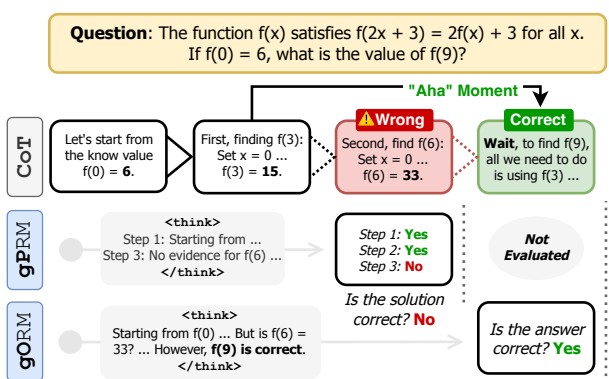

Figure 1: **Evaluating CoTs using gORM and gPRM**.

outperforms dORM, and generative variants outperform discriminative ones. In the multi-domain and specialized domain setting, however, we observe contrasting results. dORM performs on par with dPRM, gPRM is not competitive, and overall, gORM delivers **consistent and significant gains** over the others.

In §4, we identify two factors underlying the failure of gPRM. First, on more difficult multi-domain problems, LLMs tend to produce longer CoTs that PRMs struggle to evaluate. As illustrated in Fig. 1, stepwise aggregation in PRMs often fails to reward long CoTs that recover from earlier errors ("aha" moments; Guo et al., 2025), because verification stops at the first mistake. In §4.1, we analyze how this PRM-style aggregation compounds errors as the chain length increases, and confirm this effect with our empirical results. Second, label noise is prevalent in multi-domain datasets. Given that step annotation in specialized domains is costly, prior work such as Zeng et al. (2025) depends on LLM-based auto-labeling, which can introduce noise. In §4.2, we show that, under a simulated label-noise analysis in the math domain, dORM is particularly sensitive to noisy step labels, whereas gORM remains robust. Although gPRM is robust to label noise in the math domain, it degrades in the multi-domain setting. We attribute this degradation to a severe shift in the CoT-length distribution induced by consensus filtering. Based on this analysis, we present practical guidelines for selecting among the four variants and discuss limitations and future work in §5.

Our contributions and findings are summarized as follows:

- We present the first controlled multi-domain TTS evaluation of four verifier variants (dORM, dPRM, gORM, and gPRM) trained and evaluated under a common protocol across 14 diverse domains.

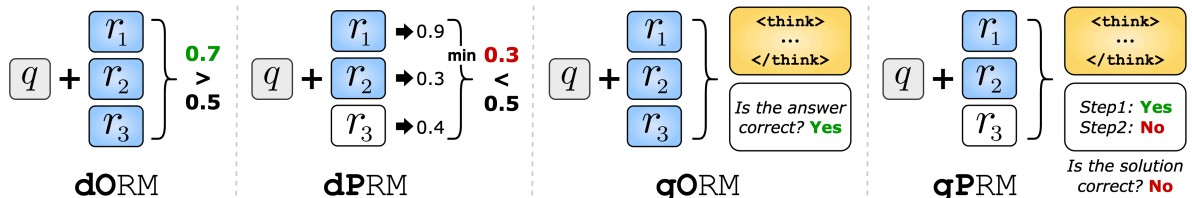

Figure 2: **Conceptual illustration of reward models**: $r_2$ is the first incorrect step; the final answer is correct.

- In contrast to conventional wisdom in math, we observe that (i) dORM performs similarly to dPRM, (ii) gPRM is not competitive, and (iii) overall, gORM **delivers consistent gains** over the others.

- To explain the empirical observations, we provide two perspectives: (i) a theoretical analysis, with empirical support, showing that *PRM risk increases with CoT length*, and (ii) evidence of *process label risk* in the multi-domain setting with length-distribution shift induced by *consensus filtering*.

## 2 Background and Related Work

In this section, we review background and related work. We first formalize notation and test-time scaling in §2.1, and then discuss reward-model variants in §2.2, summarized in Fig. 2.

### 2.1 Problem Formulation

**Notation.** For a given question $q$ with the corresponding ground-truth (GT) answer $a$, we leverage the reasoning ability of large language models (LLMs) to reliably predict $a$ by generating a chain-of-thought (CoT), *i.e.*, $r_{1:T} := (r_1, \ldots, r_T) \sim p_{\text{LLM}}(\cdot \mid q)$. Following Zeng et al. (2025), we segment the reasoning steps $r_{1:T}$ using the delimiter "\n\n", where $T$ is the number of reasoning steps. Let $x := (q, r_{1:T}) \in \mathcal{X}$, where $\mathcal{X}$ denotes the space of questions and reasoning chains, and let $x_{1:t} := (q, r_{1:t})$ be the prefix up to the $t$-th step. We consider two types of labels: (1) the *outcome label* $y = \mathbb{1}(\hat{a}(r_T) = a) \in \{0,1\}$, where $\hat{a}(r_T)$ is the predicted answer parsed from the last reasoning step $r_T$ and $\mathbb{1}$ is the indicator function; and (2) the *process labels* $z_{1:T} = (z_1, \ldots, z_T) \in \{0,1\}^T$, where each $z_t$ indicates whether the corresponding reasoning step $r_t$ is correct. Note that $y$ represents the correctness label for the last reasoning step, so $y = z_T$. With a slight abuse of notation, we also use $y$ (and $z_t$) as the corresponding random variable, where the event $y = 1$ corresponds to the model generating the positive verdict token (*e.g.*, "Yes"). Thus, expressions such as $p(y = 1 \mid x)$ denote the predicted probability of this event and are well-defined at test time without access to the true label.

**Test-time scaling (TTS) with reward models.** Reward models have many applications, including LLM training via reinforcement learning (Ziegler et al., 2019; Ouyang et al., 2022; Achiam et al., 2023; Dubey et al., 2024; Team et al., 2024; Yang et al., 2025), preference labeling (Dong et al., 2024; Meng et al., 2024; Adler et al., 2024), rejection sampling (Gulcehre et al., 2023; Dong et al., 2023), and data filtering (Dubey et al., 2024; Albalak et al., 2024; Yang et al., 2025). In this work, we focus on parallel or sampling-based (Wu et al., 2024) TTS with reward models, such as Best-of-$N$ (BoN; Charniak & Johnson, 2005; Khalifa et al., 2023; Snell et al., 2025), which allocates more compute at test time (*i.e.*, generates $N$ CoTs) and selects the candidate $\hat{a}(r_T^{(i_\star)})$ with the highest *reward*:

$$i_\star = \arg\max_{i \in \{1, \ldots, N\}} f(x^{(i)}), \quad \text{where } x^{(i)} := (q, r_{1:T}^{(i)}), \text{ and } r_{1:T}^{(i)} \overset{\text{i.i.d.}}{\sim} p_{\text{LLM}}(\cdot \mid q). \tag{1}$$

Here, $f : \mathcal{X} \to [0,1]$ is the *true (unknown) reward function* that assigns higher scores to CoTs that yield more reasonable and correct answers. However, $f$ is unknown, so we train an *external verifier* $\hat{f} : \mathcal{X} \to [0,1]$ to approximate $f$ and use $\hat{f}$ as a surrogate in Eq. (1), which is detailed in §2.2.

### 2.2 Reward Models

**Discriminative outcome reward model (dORM).** Early studies on reward models (Cobbe et al., 2021; Uesato et al., 2022; Yu et al., 2024) train a binary classifier $\hat{f}_{\text{dORM}} : \mathcal{X} \mapsto [0,1]$ on *outcome labels* $y \in \{0,1\}$

*only*, without requiring the intermediate process labels $(z_1, \ldots, z_{T-1})$. Specifically, they sample CoTs and answers for given questions, construct a training dataset $\mathcal{D}_{\mathsf{dORM}} := \{(x,y)\}$, and train $\hat{f}_{\mathsf{dORM}}$ with the binary cross-entropy (BCE) loss to approximate true $p(y = 1 \mid x)$:

$$\mathcal{L}_{\mathsf{dORM}} := \frac{1}{|\mathcal{D}_{\mathsf{dORM}}|} \sum_{(x,y) \in \mathcal{D}_{\mathsf{dORM}}} \ell_{\mathsf{BCE}}\left(\hat{f}_{\mathsf{dORM}}(x), y\right), \tag{2}$$

with $\ell_{\mathsf{BCE}}(x,y) = -\big[y \log x + (1-y) \log(1-x)\big]$. It is important to note that while dORM takes the full chain-of-thought as input and can leverage intermediate reasoning to predict final correctness, its fundamental characteristic is *outcome-level supervision and global scoring*. Despite this coarser supervision, Uesato et al. (2022) demonstrated that outcome-based feedback achieves comparable final-answer accuracy to process supervision when training the generator LLM $p_{\mathsf{LLM}}$. While their focus is on policy training rather than external test-time verification, their finding highlights the strong baseline efficacy of outcome signals.

**Discriminative process reward model (dPRM).** dPRM seeks to improve the reward signal by training on fine-grained feedback for intermediate reasoning steps, *i.e.*, *process labels* $z_{1:T}$. For dPRM, the quality of these labels is the primary factor. Accordingly, prior work has proposed collecting process labels for sampled CoTs via manual annotation (Lightman et al., 2024), Monte Carlo (MC) rollouts (Wang et al., 2024a), automatically generated labels from LLMs (Zeng et al., 2025), or combinations thereof (Zhang et al., 2025b). After collecting the process labels, we construct the training set $\mathcal{D}_{\mathsf{dPRM}} := \{(x,z_{1:T})\}$ and train $\hat{f}_{\mathsf{dPRM}}$ using the BCE loss at each step:

$$\mathcal{L}_{\mathsf{dPRM}} := \frac{1}{|\mathcal{D}_{\mathsf{dPRM}}|} \sum_{(x,z_{1:T}) \in \mathcal{D}_{\mathsf{dPRM}}} \frac{1}{T'} \sum_{t=1}^{T'} \ell_{\mathsf{BCE}}\left(\hat{f}_{\mathsf{dPRM}}(x_{1:t}), z_t\right), \tag{3}$$

where $T'$ is the index of the first incorrect reasoning step, *i.e.*, $T' := \min(\{t \in \{1,\ldots,T\} : z_t = 0\} \cup \{T\})$. Training up to the $T'$-th step reflects a common assumption in the literature (Lightman et al., 2024; Wang et al., 2024a; Zheng et al., 2024; Zeng et al., 2025): once a reasoning step is incorrect, *subsequent steps are also incorrect*, *i.e.*, if $z_t = 0$ then $z_{t'} = 0$ for all $t' \in \{t+1,\ldots,T\}$. At test time, we approximate $f$ in Eq. (1) by aggregating the step rewards with the *minimum*[1] (Zeng et al., 2025). Thus, in contrast to the global scoring of ORMs, the defining characteristic of PRMs is *process-level supervision and step-score aggregation*.

**LLM-as-a-judge.** Wang et al. (2023); Liu et al. (2023); Zheng et al. (2023) show that the task-generalization ability of LLMs can extend to verification (*i.e.*, zero-shot CoT verification). However, LLMs often "overthink" (Bavaresco et al., 2025) and, without additional training, remain practically limited (Zheng et al., 2024), implying the need for LLMs explicitly trained for verification.

**Generative outcome reward model (gORM).** Zhang et al. (2025a) proposed gORM, trained to generate a *verification CoT* together with a binary verdict, *e.g.*, "`Verification: Is the answer correct? Yes`" or "`No`". Because GT verification CoTs are unavailable, they synthesize training data via a consensus-filtering mechanism (Wang et al., 2024b; Zhu et al., 2025). We first sample a verification CoT and verdict from an LLM-as-a-judge, *i.e.*, $v_{1:L} \sim p_{\mathsf{LLM-j}}(\cdot \mid x)$ using the prompt format in Fig. 20. Here, $v_{1:L} \in \mathcal{V}^L$ denotes the verification-CoT token sequence (including the verdict tokens), $\mathcal{V}$ is the vocabulary, and let $\hat{y} \in \{0,1\}$ be the parsed verdict (1 for "`Yes`", 0 for "`No`"). We then include $(x, v_{1:L})$ in the training set $\mathcal{D}_{\mathsf{gORM}}$ only if $\hat{y}$ agrees with the known outcome label $y$. We train $p_{\mathsf{gORM}}$ with the next-token prediction over verification CoTs $v_{1:L}$:

$$\mathcal{L}_{\mathsf{gORM}} := \frac{1}{|\mathcal{D}_{\mathsf{gORM}}|} \sum_{(x,v_{1:L}) \in \mathcal{D}_{\mathsf{gORM}}} \frac{1}{L} \sum_{i=1}^{L} -\log p_{\mathsf{gORM}}(v_i \mid x, v_{<i}). \tag{4}$$

$-\log p_{\mathsf{gORM}}$ is implemented as the cross-entropy loss over $\mathcal{V}$. At test time, we approximate $f$ with:

$$\hat{f}_{\mathsf{gORM}}(x) := \mathbb{E}_{v_{1:L} \sim p_{\mathsf{gORM}}(\cdot \mid x)}\big[p_{\mathsf{gORM}}(y = 1 \mid x, v_{1:L})\big] \approx \frac{1}{M} \sum_{i=1}^{M} p_{\mathsf{gORM}}\left(y = 1 \mid x, v_{1:L}^{(i)}\right), \tag{5}$$

---

[1]Tables 8 to 12 show that minimum, average, product, and last-step aggregation yield only marginal differences, with minimum aggregation slightly outperforming the others overall, consistent with Zeng et al. (2025).

where $v_{1:L}^{(i)} \overset{\text{i.i.d.}}{\sim} p_{\text{gORM}}(\cdot \mid x)$. Here, the expectation is approximated with $M$ MC samples and the model's normalized probability of predicting the verdict "`Yes`" at the last verdict position:

$$p_{\text{gORM}}(y = 1 \mid v_{1:L}, x) := \frac{p_{\text{gORM}}(\text{``Yes''} \mid x, v_{1:(L-1)})}{p_{\text{gORM}}(\text{``Yes''} \mid x, v_{1:(L-1)}) + p_{\text{gORM}}(\text{``No''} \mid x, v_{1:(L-1)})}. \tag{6}$$

**Generative Process Reward Model (gPRM).** Beyond gORM, Khalifa et al. (2025) proposed gPRM, which is trained to generate verification CoTs $v_{1:L}$ with *stepwise process verdicts*, *e.g.*, "`Step t: The step is \boxed{correct}`" or "`\boxed{incorrect}`". Let the predicted verdict sequence be $\hat{z}_{1:T'} \in \{0,1\}^{T'}$, defined up to the first predicted incorrect step $T'^2$. Following Khalifa et al. (2025), we append a final verdict prompt, yielding the token sequence $v_{1:L^+}$ by concatenating either "`Is the solution correct? Yes`" or "`No`"—"Yes" if all predicted process labels are 1 ($\hat{z}_{1:T'} = \mathbf{1}_{T'}$), and "No" otherwise. We then construct $\mathcal{D}_{\text{gPRM}} := \{(x, v_{1:L^+})\}$ only when the predicted prefix agrees with the GT ($\hat{z}_{1:T'} = z_{1:T'}$). We train $p_{\text{gPRM}}$ with $v_{1:L^+}$:

$$\mathcal{L}_{\text{gPRM}} := \frac{1}{|\mathcal{D}_{\text{gPRM}}|} \sum_{(x, v_{1:L^+}) \in \mathcal{D}_{\text{gPRM}}} \frac{1}{L^+} \sum_{i=1}^{L^+} -\log p_{\text{gPRM}}(v_i \mid x_{1:T'}, v_{<i}). \tag{7}$$

We condition on $x_{1:T'}$ rather than the full input $x$ for training (Khalifa et al., 2025), since the model $p_{\text{gPRM}}$ is prompted to stop verification once it reaches the first incorrect step, analogous to the data-generation process (Fig. 22). At test time, consistent with Eqs. (5) and (6), we approximate $f$ in Eq. (1) by sampling from $p_{\text{gPRM}}$ and computing the normalized probability of a positive final verdict:

$$\hat{f}_{\text{gPRM}}(x) := \mathbb{E}_{v_{1:L^+} \sim p_{\text{gPRM}}(\cdot \mid x)}[p_{\text{gPRM}}(y = 1 \mid x, v_{1:L^+})] \approx \frac{1}{M} \sum_{i=1}^{M} p_{\text{gPRM}}\left(y = 1 \mid x, v_{1:L^+}^{(i)}\right), \tag{8}$$

$$p_{\text{gPRM}}(y = 1 \mid x, v_{1:L^+}) := \frac{p_{\text{gPRM}}(\text{``Yes''} \mid x, v_{1:(L^+-1)})}{p_{\text{gPRM}}(\text{``Yes''} \mid x, v_{1:(L^+-1)}) + p_{\text{gPRM}}(\text{``No''} \mid x, v_{1:(L^+-1)})}, \tag{9}$$

where $v_{1:L^+}^{(i)} \overset{\text{i.i.d.}}{\sim} p_{\text{gPRM}}(\cdot \mid x)$ and we now condition on the full input $x$ at test time (Khalifa et al., 2025). Beyond Khalifa et al. (2025), recent work has proposed more advanced gPRM architectures to improve generalization and mitigate evaluation artifacts, such as incorporating code verification (Zhao et al., 2025) or reasoning-driven generative evaluations (She et al., 2025). However, these methods are primarily optimized for the math domain and do not directly extend to multi-domain data (*e.g.*, legal or medical domains), so we follow the foundational approach of Khalifa et al. (2025) in this work.

## 3 Experiments

In this section, we evaluate dORM, dPRM, gORM, and gPRM in the math domain and the multi-domain setting. We detail experimental setups (§3.1), and present experimental results (§3.2).

### 3.1 Experimental Setups

**Math datasets.** For the math domain, we use **PRM800K** (Lightman et al., 2024) for training, where the process labels $z_{1:T}$ are human-annotated. As a testbed, we use **ProcessBench** (Zheng et al., 2024) with four splits: GSM8K, Math, Omni-Math, and OlympiadBench. We generate $N{=}16$ CoTs per question in GSM8K and Math with Qwen2.5-7B-Instruct (Team, 2024a) for TTS; since we only seek to verify that a controlled evaluation reproduces prior findings, we restrict TTS to this setting.

---

[2] As shown in Fig. 21, when generating verification CoTs for gPRM (*i.e.*, $v_{1:L} \sim p_{\text{LLM-j}}(\cdot \mid x)$), Khalifa et al. (2025) instruct the LLM-as-a-judge $p_{\text{LLM-j}}$ to stop once it detects the first incorrect step.

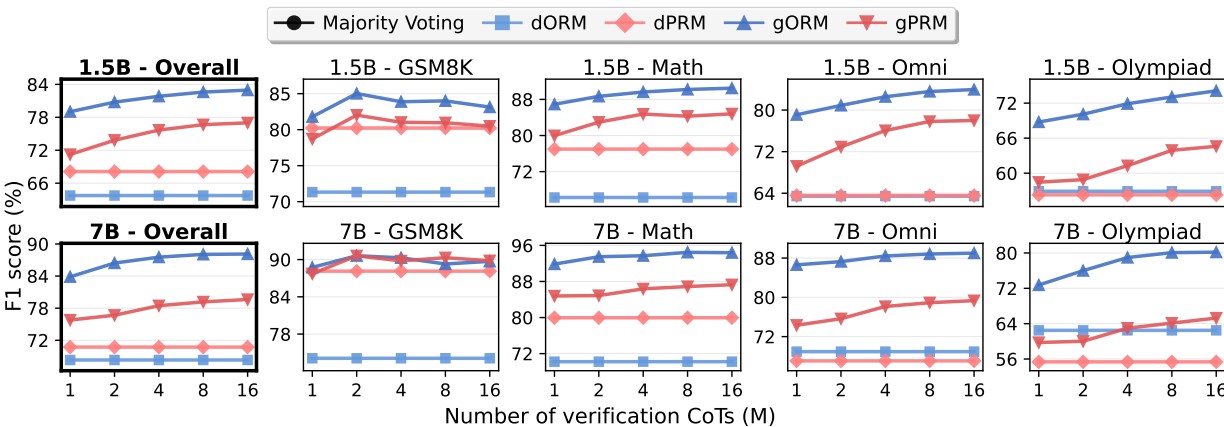

Figure 3: **Outcome verification results** on ProcessBench in the math domain.

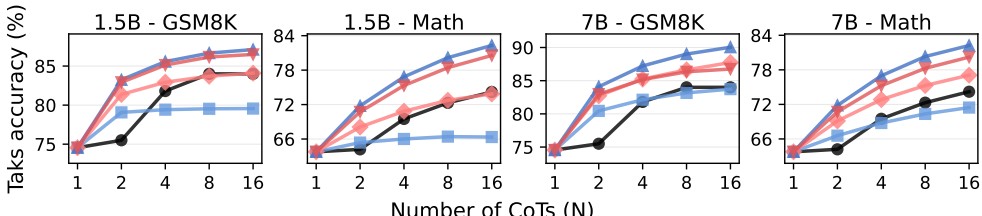

Figure 4: **Best-of-$N$ results using Qwen2.5-7B-Instruct** on GSM8K and Math in the math domain.

**Multi-domain and specialized-domain datasets.** Following Zeng et al. (2025), we mainly adopt **MMLU-Pro** (Wang et al., 2024c), a 10-choice benchmark spanning 14 domains. For training/evaluation of reward models, each question is paired with 16/128 CoTs generated by Llama-3.1-8B-Instruct (Dubey et al., 2024), where process labels $z_{1:T}$ are *automatically annotated* by Llama-3.1-70B-Instruct. To assess generalization across different $p_{\text{LLM}}$, we generate $N{=}16$ CoTs per question using SmolLM3-3B (Bakouch et al., 2025), Qwen2.5-7B-Instruct, gemma-2-9b-it (Team et al., 2024), and Llama-3.1-70B-Instruct. For broader assessment, we also include another multi-domain dataset (**GPQA-diamond**; Rein et al., 2024), as well as the medical benchmark (**MedQA**; Jin et al., 2021) and the legal benchmark (**LEXam**; Fan et al., 2026). We defer further details, including the prompts and dataset statistics, to Appendix C.

**Implementation details.** For the reward-model backbones, we use the R1-Distill models (Guo et al., 2025): Qwen-1.5B and Qwen-7B for the math domain, and Llama-8B and Qwen-14B for the multi-domain setting, respectively. We also use Qwen3-8B as the backbones to assess whether the results hold for non-distilled backbones in the multi-domain setting. For prompt templates of gORM/gPRM, we follow Zhang et al. (2025a)/Khalifa et al. (2025) (Figs. 20 and 22). We optimize reward models using AdamW (Loshchilov & Hutter, 2019) with LoRA (Hu et al., 2022). For gORM and gPRM, we sample $M{=}16/10$ verification CoTs (*cf.* Eqs. (5) and (8)) in the math/multi-domain settings, using vLLM (Kwon et al., 2023). See Appendix D and Table 2 and this repository for more details.

**Verification CoTs.** Following Zhang et al. (2025a) and Khalifa et al. (2025), we construct verification–CoT datasets for gORM and gPRM by prompting QwQ-32B (Qwen Team, 2025) with the formats in Figs. 20 and 21. We discard any verification CoT whose parsed labels are inconsistent with the targets (*e.g.*, $y$ or $z_{1:T}$), corresponding to the *consensus filtering* in §2.2. The training sets of gORM/gPRM contain 34,286/35,666 and 171,780/94,156 verification CoTs for the math and multi-domain settings. See Appendix D and Figs. 23 and 24 for more details and examples.

## 3.2 Experimental Results

**Math-domain results.** First, we evaluate the four verifier variants in the math domain. We compare outcome-verification performance with a 0.5 decision threshold, *i.e.*, $\hat{y} \coloneqq \mathbb{1}(\hat{f}(x) > 0.5)$. Fig. 3 reports F1

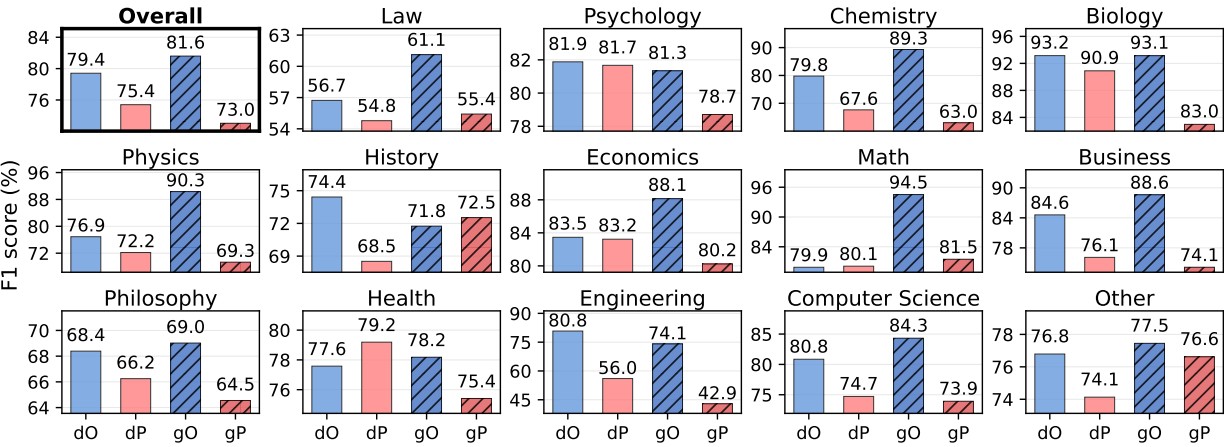

Figure 5: **Outcome verification results** on MMLU-Pro in the multi-domain setting.

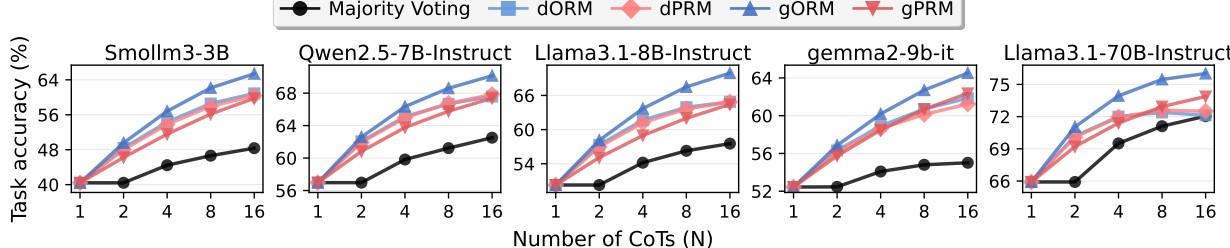

Figure 6: Overall Best-of-$N$ results using five different $p_{\text{LLM}}$ on **MMLU-Pro**.

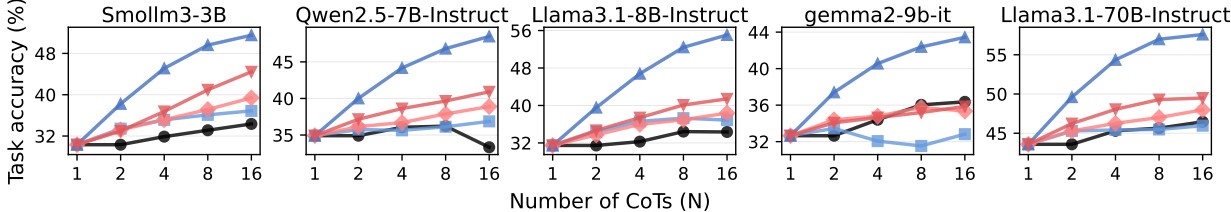

Figure 7: Best-of-$N$ results using five different $p_{\text{LLM}}$ on **GPQA-diamond**.

score (%) on ProcessBench splits. dPRM outperforms dORM overall, consistent with prior findings (Lightman et al., 2024), and shows a slight drop in Omni-Math/OlympiadBench with 7B backbones. For gORM/gPRM, the overall performance improves with $M$. At small $M$, gPRM may lag behind discriminative models (*e.g.*, OlympiadBench). gORM generally outperforms gPRM (except 7B-GSM8K), and the gap widens on Omni-Math/OlympiadBench.

Although TTS has been well studied in the math domain, evaluations are not fully controlled: (i) models are rarely compared with a shared backbone, and (ii) gORM and gPRM have not been directly compared. We therefore evaluate the reward models with BoN under controlled conditions. As shown in Fig. 4, and consistent with the findings of Lightman et al. (2024), dPRM outperforms dORM. Notably, dORM even underperforms majority voting (MV) with 1.5B backbones, demonstrating the limitations of coarse outcome-level supervision in the math domain. In line with Zhang et al. (2025a) and Khalifa et al. (2025), generative models outperform discriminative ones, with gORM slightly surpassing gPRM.

**Multi-domain and specialized-domain results.** Next, we compare the four variants in the multi-domain setting. Fig. 5 reports F1 scores (%) for outcome-verification, with a 0.5 decision threshold, using R1-Distill-Qwen-14B as the reward model backbone. dO/dP/gO/gP denote dORM/dPRM/gORM/gPRM. In contrast to the math domain results in Fig. 3, ORM variants achieve higher F1 scores than PRM variants.

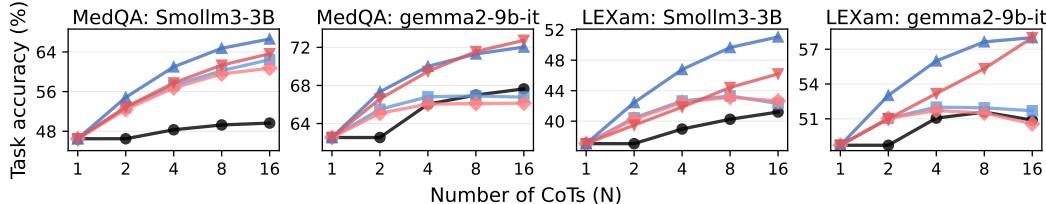

Figure 8: Best-of-$N$ results using two different $p_{\text{LLM}}$ on **MedQA** and **LEXam**.

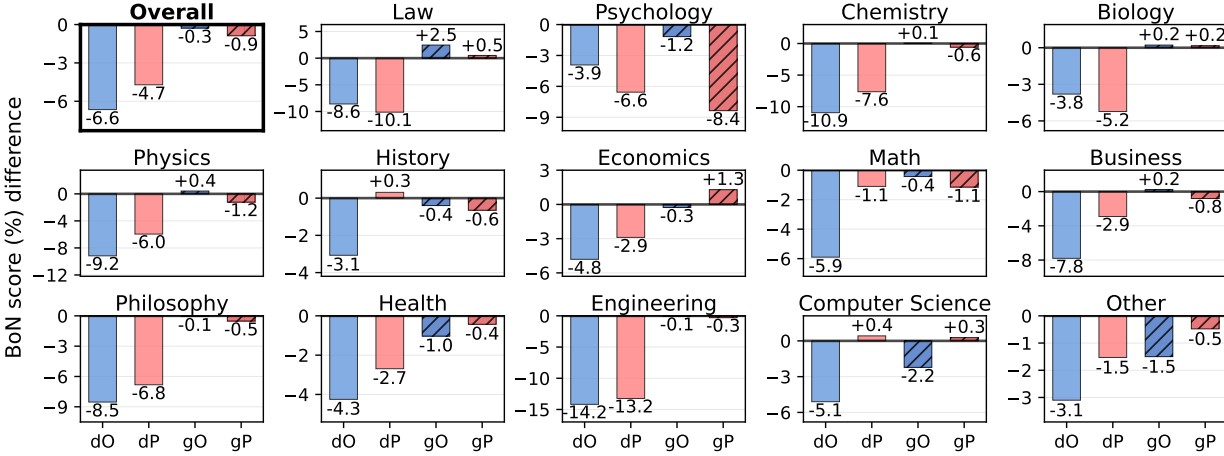

Figure 9: **Best-of-$N$ performance gap between all-domain and single-domain training** on MMLU-pro.

Fig. 6 shows the overall Bo$N$ performance using five different $p_{\text{LLM}}$ and R1-Distill-Qwen-14B as the reward model backbone. In this setting, dORM *performs comparably* to dPRM, while gPRM is *not competitive*, which is contrary to Lightman et al. (2024); Khalifa et al. (2025) and our math-domain results in Fig. 4. Overall, gORM outperforms dORM/dPRM/gPRM, without notable degradation in any domain relative to the others (see Appendix F for detailed per-domain results). The same pattern holds for a smaller backbone (DeepSeek-R1-Distill-Llama-8B; Fig. 27) and a non-distilled backbone (Qwen3-8B; Fig. 39). Table 3 also shows that this trend is not due to insufficient hyperparameter tuning: sweeping the learning rate and LoRA rank yields only marginal changes in PRM performance.

To verify that the above observations generalize across datasets, we take the reward models trained on the MMLU-Pro training split and evaluate them on GPQA-diamond, MedQA, and LEXam by generating $N{=}16$ CoTs for each question. For MedQA and LEXam, we include only SmolLM3-3B and gemma-2-9b-it, since the other $p_{\text{LLM}}$ exhibit severe degradation, even compared to random guessing. As shown in Fig. 7, gORM outperforms dORM/dPRM/gPRM on GPQA-diamond, consistent with the results on MMLU-Pro (Fig. 6). In specialized domains (Fig. 8), generative variants significantly outperform discriminative variants; gORM outperforms gPRM on CoTs generated by SmolLM3-3B and performs comparably on CoTs generated by gemma-2-9b-it.

**Effect of multi-domain training.** To assess the effect of multi-domain training, we train and evaluate all four reward model variants *only* on each MMLU-Pro domain and compare each variant to its multi-domain counterpart. Fig. 9 reports the degradation in Bo$N$ performance with $N{=}16$ under domain-specialized training. We observe *severe drops* for dORM and dPRM, with a slightly larger decline for dORM, likely because outcome-only supervision is sparser than step-level supervision and both are relatively data-hungry. In contrast, gORM and gPRM appear more *sample-efficient*: even without multi-domain training, their performance decreases only modestly (or in some cases improves), demonstrating the efficiency of generative reward models. This also explains the results on MedQA and LEXam (Fig. 8): the generative variants show strong gains over the discriminative variants in these data-hungry specialized domains (medical and legal). We defer complete results of single-domain training for four reward models to Figs. 37 and 38 in Appendix F.

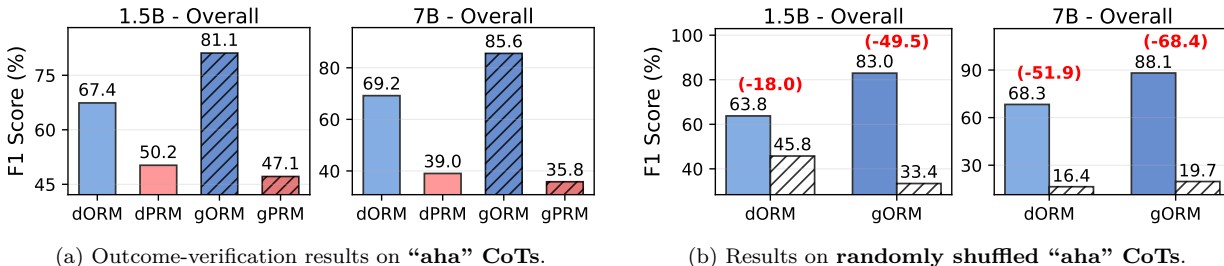

(a) Outcome-verification results on **"aha" CoTs**.     (b) Results on **randomly shuffled "aha" CoTs**.

Figure 10: **(a)**: ORMs outperform PRMs on "aha" CoTs; however, **(b)**: their performance drops when intermediate steps are randomly shuffled. This suggests that ORMs **do not simply memorize question-answer pairs**.

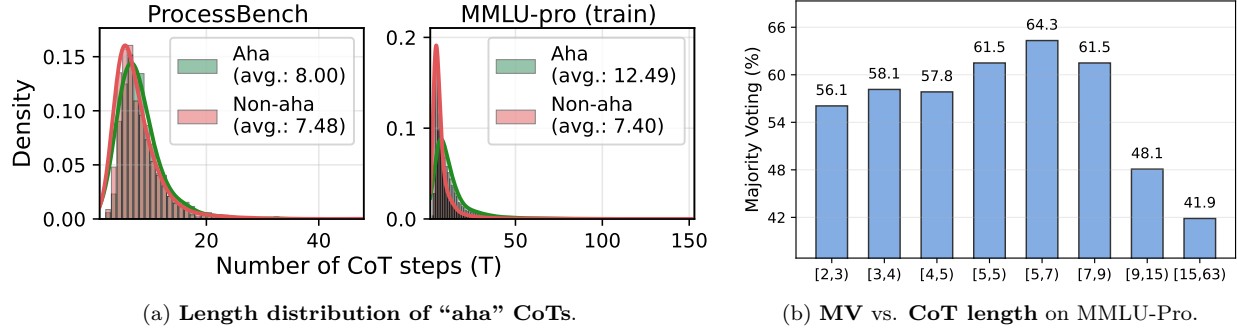

(a) **Length distribution of "aha" CoTs**.     (b) **MV** vs. **CoT length** on MMLU-Pro.

Figure 11: **(a)**: "Aha" moments lengthen CoTs, an effect pronounced in the multi-domain setting (MMLU-Pro); and **(b)**: majority voting results degrade significantly with increasing CoT length.

## 4 Analysis on Why PRMs Fail in the Multi-Domain Settings

In this section, we analyze the failure modes of PRMs observed in the multi-domain setting of §3.

### 4.1 Risk of PRMs with CoT Length

**"Aha" CoTs.** As noted in §2.2, PRMs typically assume that once a reasoning step is incorrect, *all subsequent steps are incorrect*. However, recent reasoning models can recover from earlier mistakes and still arrive at the correct answer (an **"aha" moment**; Guo et al., 2025). In such cases, PRMs can miss the recovery due to a monotonicity bias induced by their training data. To demonstrate this, we evaluate on "aha" CoTs from ProcessBench[3] that contain at least one incorrect step ($\exists\, t \in \{1, \ldots, T\} : z_t = 0$) but a correct outcome ($y = 1$). Overall, "aha" CoTs account for 15.3% of the cases. In Fig. 10a, we report F1 scores (%) for the "aha" CoTs using $M{=}16$ for gORM/gPRM. We observe that PRM variants perform particularly **poorly on "aha" CoTs**. Moreover, scaling the backbone from 1.5B to 7B improves ORM performance, whereas PRM performance degrades with larger backbones, possibly because larger PRMs are more likely to follow the PRM assumption inherent in their training data and objective (*cf.* Eqs. (3) and (7)).

**Do ORMs overfit on "aha" CoTs?** A natural concern about ORM results on "aha" CoTs in Fig. 10a is *overfitting*: ORMs might only memorize questions and their answers, thereby correctly verifying "aha" CoTs without checking the correctness of intermediate reasoning steps. This memorization issue in the math domain has recently been studied by Wu et al. (2025). To investigate this, we conduct the following test: (i) **replace the intermediate reasoning steps** $r_{1:T-1}$ with $r'_{1:T-1}$ **taken from other CoTs**, and (ii) evaluate ORMs on these perturbed CoTs. If ORMs only memorize the answer in the final reasoning step $r_T$, their performance should remain largely unaffected. However, Fig. 10b shows a **significant drop** for ORMs (dashed), indicating the reliance on intermediate steps. Interestingly, the degradation is greater with

---

[3]The "aha" experiments are conducted only on ProcessBench, which provides human-annotated step-level labels. For MMLU-Pro, LLM-annotated process labels would make the rate of "aha" CoTs depend on annotator reliability.

the 7B backbone than with the 1.5B backbone for both **dORM** and **gORM**. This suggests that larger models rely *more heavily* on intermediate reasoning steps during verification.

**Risk increases with CoT length.** "Aha" moments can also lengthen CoTs, an effect especially pronounced in the multi-domain setting (Fig. 11a), where LLMs struggle more than in math. As shown in Fig. 11b, majority voting results degrade significantly with increasing CoT length in the multi-domain setting. Consistent with the outcome-verification failures of PRMs on "aha" CoTs, we argue that *the error of PRM variants grows with CoT length* ($T$). Intuitively, as a CoT grows longer, the chance that a PRM misclassifies at least one intermediate step rises, making it more likely to *prematurely* conclude the CoT is incorrect. Longer CoTs also create more opportunities for "aha" recoveries that PRMs systematically miss. We formalize this as follows:

**Theorem 4.1** (Informal: Log-error bound of **dORM** and **gORM**; Theorem A.1)**.** *Under mild assumptions on the variance of the reward model's error, the expected squared log-error of **dORM** and **gORM** is bounded by a constant that is **independent** of the CoT length $T$.*

**Theorem 4.2** (Informal: Log-error lower bound of **dPRM**; Theorem A.2)**.** *Assuming a minimum average per-step error variance and bounded local error correlation, the expected squared log-error of **dPRM** grows **at least linearly** with the CoT length $T$.*

**Theorem 4.3** (Informal: Log-error lower bound of sampled **gPRM**; Theorem A.3)**.** *Under the same assumptions as Theorem 4.2, the expected squared log-error of **gPRM** also grows **at least linearly** with $T$. Furthermore, the generative sampling process introduces additional per-step variance, which strictly worsens this lower bound compared to **dPRM**.*

All formal definitions, assumptions, and proofs are deferred to Appendix A. Theorem 4.1 to 4.3 establish that while **dORM/gORM** error bounds remain stable regardless of length, **dPRM/gPRM** lower bounds grow linearly with $T$. In Theorem A.4, we additionally show that for **gPRM** with Monte Carlo estimation (*cf.* Eq. (8)), the log-error lower bound also increases linearly with $T$.

**Empirical support.** To empirically support Theorem 4.1 to 4.3, we plot the F1 score (%) for outcome-verification in the multi-domain setting as a function of the number of reasoning CoT steps ($T$) in Fig. 12-(Left). We divide CoTs into eight bins: 1-2, 3, 4, 5, 6, 7-8, 9-11, and 14-162 steps. As $T$ increases, **dPRM/gPRM** degrade considerably relative to **dORM/gORM**. Fig. 12-(Right) shows the performance improvements over majority voting for different categories with respect to the average number of CoT steps. We observe negative correlations for **dPRM** (-0.058) and **gPRM**

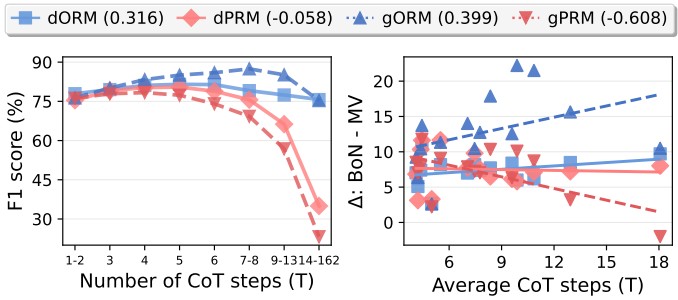

Figure 12: **(Left)**: Outcome verification vs. CoT length; **(Right)**: TTS improvement vs. average CoT length.

(-0.608), while **dORM** (0.316) and **gORM** (0.399) show positive correlations. These results not only provide empirical support for Theorem 4.1 to 4.3 but also demonstrate that increasing CoT length can degrade TTS performance for **dPRM** and **gPRM** in the multi-domain setting.

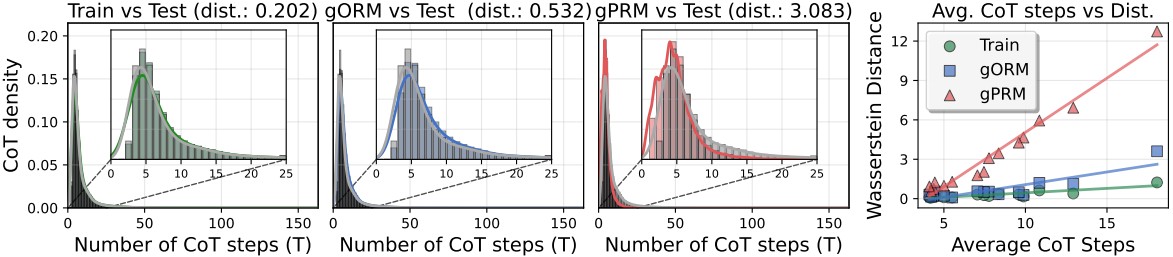

Figure 14: **Length distribution shift** on MMLU-Pro (overall/per-domain) measured by Wasserstein distance.

## 4.2 Label Noise and Consensus Filtering of PRMs

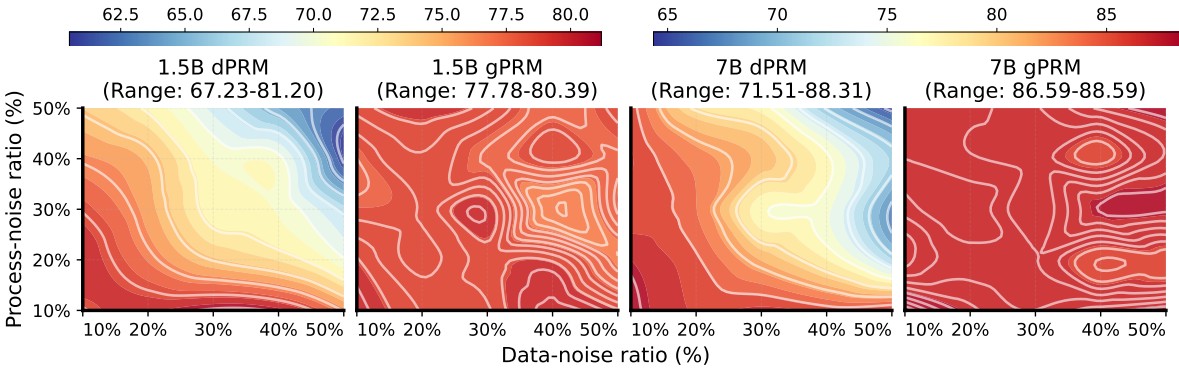

Figure 13: **Outcome-verification results of PRMs vs. label noise** on GSM8K.

**Label noise risk.** Beyond CoT-length effects, *label noise* poses an additional risk, especially in multi-domain settings. Since human annotation of long CoTs is more costly in specialized domains such as law and medicine than in math, prior work often relies on LLMs to auto-label process steps (Zeng et al., 2025), which introduces noise that can degrade PRM performance. Indeed, recent evaluations in the math domain document that automated annotation pipelines, such as Monte Carlo estimation, can inject substantial label noise and evaluation artifacts (Zhang et al., 2025b). We study this by injecting synthetic noise into the process labels of PRM800K. We study this by injecting synthetic noise into the process labels of PRM800K. We vary the level of noise along two axes: (i) *process-noise ratio* (the per-step probability of flipping a process label) and (ii) *data-noise ratio* (the fraction of examples to which noise is applied). We report the outcome-verification F1 score (%) in Fig. 13 using 1.5B backbones and 7B backbones, using greedy decoding for generative variants. dPRM is **highly sensitive** to label noise, demonstrating its potential vulnerability in multi-domain settings. In contrast, gPRM is more robust, which is consistent with reports that LLM memorization can make random label noise act as a mild regularizer in math (Wu et al., 2025).

**Length shift hurts gPRM.** We further analyze why gPRM degrades in the multi-domain setting, despite its robustness to label noise in math. As CoTs become longer, the imperfect $p_{\texttt{LLM-j}}$ struggles to align stepwise verification rationales with process labels. As a result, *consensus filtering* **prunes long CoTs**, shifting the training CoT-length distribution away from the test set (Fig. 14).

We quantify the above length distribution shift with the Wasserstein distance (Kantorovich, 1960), reporting distances from the test set to the unfiltered pool (**Train**), the gORM training set, and the gPRM training set. In the math domain (Table 4), gPRM has the smallest distance (*e.g.*, overall: 2.760/2.430/1.600 for Train/gORM/gPRM), whereas in the multi-domain setting (Fig. 14 and Table 5) it has **the largest** distance (*e.g.*, overall: 0.202/0.532/3.083 for Train/gORM/gPRM).

The distribution shift of gPRM also corresponds to its degradation across domains, observed in the multi-domain setting (Fig. 6). Fig. 15 shows a strong negative correlation between the Wasserstein distance and per-domain improvement over majority voting with $N$=16 for gPRM ($-0.625$), whereas correlations are weak for the other methods (0.345/0.048/0.082 for dORM/dPRM/gORM). Together, these results suggest

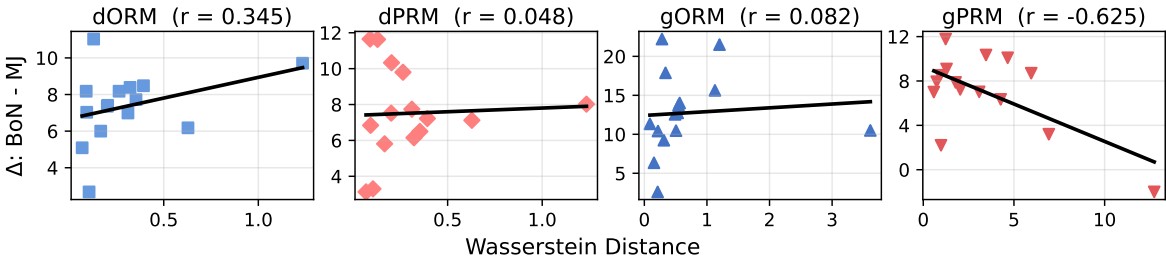

Figure 15: **Per-domain Bo$N$ improvement over majority voting** vs. **Wasserstein distance** on MMLU-Pro.

that *consensus filtering* induces a length-distribution shift that disproportionately affects gPRM in the multi-domain setting, despite its robustness to label noise.

**Label refinement and relaxed filtering.** To mitigate the CoT-length distribution shift for gPRM, we further test label refinement with Gemini-2.0 Flash (Comanici et al., 2025) and relaxed *consensus filtering*. Although both increase the CoT survival rate, they neither substantially reduce the Wasserstein distance nor improve downstream performance. See Tables 5 to 7 in Appendix G for more details.

## 5 Practical Guidelines, Limitations, and Future Work

This section suggests practical guidance, clarifies limitations, and outlines future directions. Building on the empirical and theoretical results of §3 and §4, we summarize when each reward-model variant is preferable in the table below, and then connect the entries to the supporting evidence.

| | | |
|---|---|---|
| (i) | *Short* CoTs, *clean* labels, *tight* latency | dPRM |
| (ii) | *Long* CoTs / *frequent error recoveries* | gORM if compute permits; else dORM |
| (iii) | *Mixed/shifting domains* | gORM |
| (iv) | *High label noise* | ORM
PRMs amplify early errors |
| (v) | *Strict compute/latency* | dORM/dPRM
gORM and gPRM add sampling overhead |
| (vi) | *Limited training data* | gORM/gPRM
Higher sample efficiency |

**From findings to recommendations.** Rows (i) and (ii) follow from our CoT-length analysis in §4.1. When CoTs are short and process labels are clean, the per-step signal exploited by dPRM is informative and its discriminative head keeps inference cheap, matching the math-domain regime where dPRM is competitive (Fig. 6). As $T$ grows, however, Theorem 4.2 and 4.3 show that the log-error lower bounds of dPRM and gPRM grow linearly with $T$, while Theorem 4.1 bounds the ORM error independently of $T$. This is corroborated in Fig. 12 (length-vs-improvement correlations of $-0.058/-0.608$ for dPRM/gPRM versus $0.316/0.399$ for dORM/gORM) and by the "aha" analysis in Fig. 10a, where PRMs systematically miss recoveries from earlier mistakes, hence the ORM recommendation in row (ii).

Rows (iii)–(vi) reflect the multi-domain and resource trade-offs surfaced in §3 and §4.2. For mixed or shifting domains, *consensus filtering* prunes long CoTs and induces a length-distribution shift that disproportionately hurts gPRM (largest Wasserstein distance in Table 5, with a correlation of $-0.625$ between distance and per-domain improvement in Fig. 15), motivating gORM as the default. Under high label noise, the PRM800K noise-injection study (Fig. 13) shows that dPRM is highly sensitive to per-step label flips and that even the more robust gPRM inherits the monotonicity assumption that amplifies early errors, so ORMs are the safer choice.

Finally, gORM/gPRM are inherently slower than discriminative variants because they autoregressively generate verification tokens with $M$ rollouts (*cf.* Eq. (8)), while dORM/dPRM score each CoT in a single

forward pass. On MMLU-Pro with CoTs from Llama-3.1-8B-Instruct, the normalized inference times are 0.0005/0.0005/0.0571/0.0306 seconds per CoT for dORM/dPRM/gORM/gPRM, respectively, making dORM/dPRM preferable under strict compute or latency budgets. Although gPRM is 1.87× faster than gORM due to early stopping, it may miss "aha" CoTs; thus, we recommend gORM when reliability is prioritized, especially in medical and legal domains. In contrast, generative variants better exploit LLM priors and rationale-style supervision, improving sample efficiency with limited training data (Fig. 6 and Appendix G), so we recommend them in row (vi).

**Limitations and future work.** While we present a thorough analysis of four reward model variants, our study has several limitations: (i) All models are trained via **supervised fine–tuning**. One could instead use a generative verifier to roll out rationales and treat agreement between their verdict and the GT label as a reward signal for reinforcement learning (RL). Because using RL to train verifiers/reward models is uncommon and introduces additional confounders, we exclude RL-based training from our analysis. (ii) Owing to computational constraints, we adopt **LoRA adapters** rather than full-parameter fine-tuning. This choice may affect performance and scaling behavior, however, we expect the qualitative trends to hold. (iii) Following most of the PRM literature (Lightman et al., 2024; Zeng et al., 2025), we do not consider **tool use**, however, Gou et al. (2024) showed that tool use can help reduce auto-label noise. In future work, we plan to extend our analysis to broader task domains, model families, and training regimes. We also plan to explicitly study tool-augmented verification and inference pipelines.

## Acknowledgement

This work was supported by Institute for Information & communications Technology Planning & Evaluation (IITP) grant funded by the Korea government (MSIT) (RS-2019-II190075, Artificial Intelligence Graduate School Program (KAIST)) and Center for Applied Research in Artificial Intelligence (CARAI) grant funded by DAPA and ADD (UD190031RD)

## Broader Impact Statement

This work evaluates verification strategies for test-time scaling of LLMs across multiple domains. It **does not** involve human subjects, user studies, or the collection of personally identifiable information. All datasets used are **publicly available** benchmarks and were accessed under their respective licenses. To the best of our knowledge, they do not contain sensitive personal data.

A natural direction for future work is to increase the trustworthiness of LLM outputs in real systems by verifying them, thereby reducing reasoning errors and hallucinations. Although our experiments include legal and medical-themed datasets (*e.g.*, law and health), the models and methods are research artifacts and are **not** intended for real-world legal, medical, or other high-stakes decision-making. They should not substitute professional judgment, and any deployment in such settings would require additional domain-specific validation, safety auditing, and regulatory compliance.

## Reproducibility Statement

For reproducibility, we believe that we provide sufficient materials, including prompts, hyperparameters, model backbones, training details, and the synthetic data generation process, throughout the main paper (§2 and §3.1). Additional details are deferred to Appendices B–E due to space constraints. Upon acceptance, we commit to **publicly releasing** all relevant artifacts for reproducibility: **(i) code**, **(ii) datasets** (including any we generate), and **(iii) model checkpoints**.

In the meantime, anonymized artifacts are available as follows:

Code.

- `discriminative/`: training/inference code for *discriminative* variants (dORM/dPRM), adapted from VersaPRM (Zeng et al., 2025).
- `generative/`: training/inference code for *generative* variants (gORM/gPRM).

**Training datasets.**

- `MMLU-Pro_Llama-3.1-8B-Instruct_train`: MMLU-Pro training dataset for dORM/dPRM, adapted from VersaPRM (Zeng et al., 2025).
- `MMLU-Pro_Llama-3.1-8B-Instruct_gORM_train`: MMLU-Pro training dataset for gORM.
- `MMLU-Pro_Llama-3.1-8B-Instruct_gPRM_train`: MMLU-Pro training dataset for gPRM.

**Test datasets.**

- `MMLU-Pro_Llama-3.1-8B-Instruct_test`: MMLU-Pro test dataset generated by Llama-3.1-8B-Instruct, adapted from VersaPRM (Zeng et al., 2025).
- `MMLU-Pro_SmolLM3-3B_test`: MMLU-Pro test dataset generated by SmolLM3-3B.
- `MMLU-Pro_Qwen2.5-7B-Instruct_test`: MMLU-Pro test dataset generated by Qwen2.5-7B-Instruct.
- `MMLU-Pro_gemma-2-9b-it_test`: MMLU-Pro test dataset generated by gemma-2-9b-it.
- `MMLU-Pro_Llama-3.1-70B-Instruct_test`: MMLU-Pro test dataset generated by Llama-3.1-70B-Instruct.
- `GPQA-diamond_Llama-3.1-8B-Instruct_test`: GPQA-diamond test dataset generated by Llama-3.1-8B-Instruct.
- `GPQA-diamond_SmolLM3-3B_test`: GPQA-diamond test dataset generated by SmolLM3-3B.
- `GPQA-diamond_Qwen2.5-7B-Instruct_test`: GPQA-diamond test dataset generated by Qwen2.5-7B-Instruct.
- `GPQA-diamond_gemma-2-9b-it_test`: GPQA-diamond test dataset generated by gemma-2-9b-it.
- `GPQA-diamond_Llama-3.1-70B-Instruct_test`: GPQA-diamond test dataset generated by Llama-3.1-70B-Instruct.
- `MedQA_SmolLM3-3B_test`: MedQA test dataset generated by SmolLM3-3B.
- `MedQA_gemma-2-9b-it_test`: MedQA test dataset generated by gemma-2-9b-it.
- `LEXam_SmolLM3-3B_test`: LEXam test dataset generated by SmolLM3-3B.
- `LEXam_gemma-2-9b-it_test`: LEXam test dataset generated by gemma-2-9b-it.

**Model checkpoints.**

- `dORM-14B`: dORM with DeepSeek-R1-Distill-Qwen-14B backbone.
- `dPRM-14B`: dPRM with DeepSeek-R1-Distill-Qwen-14B backbone.
- `gORM-14B`: gORM with DeepSeek-R1-Distill-Qwen-14B backbone.
- `gPRM-14B`: gPRM with DeepSeek-R1-Distill-Qwen-14B backbone.
- `dORM-8B`: dORM with DeepSeek-R1-Distill-Llama-8B backbone.
- `dPRM-8B`: dPRM with DeepSeek-R1-Distill-Llama-8B backbone.
- `gORM-8B`: gORM with DeepSeek-R1-Distill-Llama-8B backbone.
- `gPRM-8B`: gPRM with DeepSeek-R1-Distill-Llama-8B backbone.
- `dORM-qwen`: dORM with Qwen3-8B backbone.
- `dPRM-qwen`: dORM with Qwen3-8B backbone.
- `gORM-qwen`: gORM with Qwen3-8B backbone.

- `gPRM-qwen`: gPRM with Qwen3-8B backbone.

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

## Appendix Overview

This appendix provides supplementary materials to support the main paper as follows:

- **Theoretical Analysis** (Appendix A): details notations, assumptions, and proofs for Theorem 4.1 to 4.3.
- **Prompts** (Appendix B): presents the detailed prompt formats.
- **Datasets** (Appendix C): describes the datasets used in our experiments.
- **Implementation Details** (Appendix D): provides implementation details, such as (i) backbones for reward models, (ii) hyperparameters, and (iii) verification CoTs for gORM and gPRM.
- **Training Examples** (Appendix E): contains training examples including verification CoTs of gORM and gPRM in the law domain of MMLU-pro.
- **Detailed Results on MMLU-Pro** (Appendix F): includes the complete results of §3.2 (omitted in the main paper due to the space limit), such as per-domain results on MMLU-pro using weighted majority voting.
- **Additional Analysis** (Appendix G): includes the complete results of §4.

## A Theoretical Analysis

### A.1 Analysis on Log-Error Bound

**Notation.** We assume that a correct final step, $y = z_T = 1$, implies all previous steps are correct. Define the stepwise conditional probabilities $u_t := \Pr(z_t = 1 \mid x, z_1 = 1, \ldots, z_{t-1} = 1)$ for $t \in [T]$. By the chain rule and the assumption, the true reward function,

$$f(x) = p(y = 1 \mid x) = p(z_T = 1 \mid x) = p(z_{1:T} = 1 \mid x) = \prod_{t=1}^{T} u_t(x)$$

and we write $\zeta(x) := \log f(x) = \sum_{t=1}^{T} \log u_t(x)$. For dPRM, we define the stepwise conditional distribution $\hat{u}_t(x) := \hat{f}_{\text{dPRM}}(x_{1:t})$ and use product for the aggregation, *i.e.*, $\hat{f}_{\text{dPRM}}(x) := \prod_{t=1}^{T} \hat{u}_t(x)$. Similarly, we define the conditional distribution $F_t(x, v_{\leq t}) \in [0,1]$ to be the gPRM's normalized probability that step $t$ is correct given the verification prefix, *i.e.* $\hat{f}_{\text{gPRM}}(x) := \mathbb{E}_{v_{1:L^+}}[\prod_{t=1}^{T} F_t(x, v_{\leq t})]$. To bound log probability, we assume there is $\varsigma \in (0, 1/2)$ such that all probabilities/predictors appearing inside logarithms are clipped into $[\varsigma, 1-\varsigma]$. Hence all logs are finite and $|\log(\cdot)| \leq \log(1/\varsigma)$.

**Error terms.**

1. dPRM: Define $\delta_t := \log \hat{u}_t - \log u_t$ (evaluated at the appropriate prefixes), and

$$m_t := \mathbb{E}[\delta_t \mid x], \quad \xi_t := \delta_t - m_t,$$

   so $\mathbb{E}[\xi_t \mid x] = 0$.

2. dORM or gORM: Let $\epsilon \in \{\epsilon_d, \epsilon_g\}$,

$$\epsilon_d := \log \hat{f}_{\text{dORM}}(x) - \log f(x), \quad \epsilon_g := \log \hat{f}_{\text{gORM}}(x) - \log f(x),$$

   and decompose

$$\bar{m} := \mathbb{E}[\epsilon \mid x], \quad \bar{\xi} := \epsilon - \bar{m}, \quad \beta_{\text{orm}}^2 := \mathbb{E}[\bar{m}^2],$$

   so that $\mathbb{E}[\bar{\xi} \mid x] = 0$.

3. gPRM: For a single rollout $v_{1:L^+} \sim p_{\text{gPRM}}(\cdot \mid x)$, define

$$\tilde{u}_t := F_t(x, v_{\leq t}), \quad \tilde{f}_{\text{gPRM}}(x) := \prod_{t=1}^{T} \tilde{u}_t.$$

The sampled gPRM log-error is

$$\Delta_{\mathsf{gPRM}} \;:=\; \log \tilde{f}_{\mathsf{gPRM}}(x) - \zeta(x) = \sum_{t=1}^{T} \delta_t^{(g)}, \quad \delta_t^{(g)} := \log \tilde{u}_t - \log u_t.$$

Let

$$m_t^{(g)} := \mathbb{E}[\delta_t^{(g)} \mid x], \quad \xi_t^{(g)} := \delta_t^{(g)} - m_t^{(g)},$$

so that $\mathbb{E}[\xi_t^{(g)} \mid x] = 0$.

**Assumptions.** There exist constants $\sigma^2 > 0$, $\rho \geq 0$, and an integer $k \geq 0$ (independent of $T$) such that for all $x$,

1. (Average variance floors) $\frac{1}{T}\sum_{t=1}^{T} \mathrm{Var}(\xi_t \mid x) \geq \sigma^2$, $\frac{1}{T}\sum_{t=1}^{T} \mathrm{Var}(\xi_t^{(g)} \mid x) \;\geq\; \sigma^2 + \tau^2$.
2. (Local error dependence) For any step $s, t$, errors are only correlated within a local context window $k$, such that $\mathrm{Cov}(\xi_s, \xi_t \mid x) = 0$ for $|s - t| > k$. Within this window, the anti-correlation is bounded by $\mathrm{Cov}(\xi_s, \xi_t \mid x) \geq -\rho$. (The same holds for $\xi^{(g)}$).
3. (Positive slope) $\sigma^2 > 2k\rho$.

Unlike a strict per-step floor, the average variance condition naturally accommodates deterministic or purely algebraic reasoning steps (where variance is near zero), provided the cumulative chain injects a proportional amount of noise. Similarly, the local dependence assumption reflects bounded memory in autoregressive generation. While a model may self-correct recent errors within a window $k$, distant steps become conditionally independent. Notice that bounded local dependence implies $\sum_{1 \leq s < t \leq T} \mathrm{Cov}(\xi_s, \xi_t \mid x) \geq -k\rho T$.

For gPRM with *sampled* verification CoTs, sampling contributes per-step noise: $\frac{1}{T}\sum_{t=1}^{T} \mathrm{Var}(\xi_t^{(g)} \mid x) \geq \sigma^2 + \tau^2$ for some $\tau^2 > 0$. For ORMs, assume $\mathrm{Var}(\bar{\xi} \mid x) \leq \tau_{\mathrm{orm}}^2 < \infty$ (no $T$-dependence).

**Theorem A.1** (Log-error bound of dORM or gORM). *Let $\epsilon \in \{\epsilon_d, \epsilon_g\}$ and write $\epsilon = \bar{m} + \bar{\xi}$ with $\mathbb{E}[\bar{\xi} \mid x] = 0$. If $\mathrm{Var}(\bar{\xi} \mid x) \leq \tau_{\mathrm{orm}}^2$ (independent of $T$), then*

$$\mathbb{E}[\epsilon^2] \;=\; \mathbb{E}[\mathrm{Var}(\bar{\xi} \mid x)] \;+\; \mathbb{E}[\bar{m}^2] \;\leq\; \tau_{\mathrm{orm}}^2 \;+\; \beta_{\mathrm{orm}}^2,$$

*a bound that does not depend on the CoT length $T$.*

**Theorem A.2** (Log-error lower bound of dPRM). *Let $\Delta_{dPRM} := \log \hat{f}_{dPRM}(x) - \zeta(x)$. Under the assumptions above,*

$$\mathbb{E}[\Delta_{dPRM}^2] \;\geq\; (\sigma^2 - 2k\rho)T.$$

**Theorem A.3** (Log-error lower bound of gPRM). *Under the assumptions above,*

$$\mathbb{E}[\Delta_{gPRM}^2] \;\geq\; (\sigma^2 + \tau^2 - 2k\rho)T.$$

**Jensen-gap representation (mean predictor).** Let $L(x,v) := \sum_{t=1}^{T} \log F_t(x, v_{\leq t})$ and $K_x(\theta) := \log \mathbb{E}[e^{\theta L} \mid x]$. Define the mean predictor $\mu(x) := \mathbb{E}[e^L \mid x]$ and $\Delta_{\mathrm{mean}}(x) := \log \mu(x) - \zeta(x)$. Then with $B^{(g)}(x) := \mathbb{E}[L \mid x] - \zeta(x)$, we have the exact decomposition

$$\Delta_{\mathrm{mean}}(x) = B^{(g)}(x) + \delta_J(x), \quad \delta_J(x) = K_x(1) - K_x'(0) = \int_0^1 (1-\theta)\,\mathrm{Var}_\theta(L \mid x)\,d\theta \geq 0,$$

where $\mathrm{Var}_\theta$ denotes variance under the exponentially tilted law $d\mathbb{P}_\theta \propto e^{\theta L} d\mathbb{P}$, i.e., $d\mathbb{P}_\theta(v) = \mathbb{1}\{M(\theta) > 0\}\, e^{\theta L(x,v)} M(\theta)^{-1} d\mathbb{P}(v)$ with $M(\theta) := \mathbb{E}[e^{\theta L} \mid x]$.

**Theorem A.4** (Log-error lower bound of mean-gPRM). *Assume the conditions of Theorem A.3. In addition, suppose there exists $\kappa \in (0,1]$ such that for all $\theta \in [0,1]$,*

$$\mathrm{Var}_\theta(L \mid x) \;\geq\; \kappa\,\mathrm{Var}(L \mid x).$$

*Then, for every $x$,*

$$\Delta_{\mathrm{mean}}(x) \ \geq \ B^{(g)}(x) \ + \ \frac{\kappa}{2}\,\mathrm{Var}(L \mid x) \ \geq \ B^{(g)}(x) \ + \ \frac{\kappa}{2}\Big((\sigma^2 + \tau^2 - 2k\rho)T\Big).$$

*Consequently,*

$$\mathbb{E}[\Delta_{\mathrm{mean}}] \ \geq \ \frac{\kappa}{2}\Big((\sigma^2 + \tau^2 - 2k\rho)T\Big) \ - \ \sqrt{\mathbb{E}[B^{(g)}(x)^2]}, \quad \mathbb{E}[\Delta^2_{\mathrm{mean}}] \ \geq \ (\max\{0, \mathbb{E}[\Delta_{\mathrm{mean}}]\})^2 \,.$$

**Takeaways.** Under mild anti-correlation and variance-floor assumptions, dPRM and sampled gPRM incur log-error that grows at least linearly in the CoT length $T$, and the additional sampling noise $\tau^2$ makes gPRM strictly worse. In contrast, ORM estimators admit error bounds that are independent of $T$ provided the conditional noise is bounded, which makes them preferable for long CoTs. For mean-gPRM, the Jensen gap introduces a strictly nonnegative bias that scales with the variance of $L$ and hence with $T$, so even a calibrated predictor ($B^{(g)} = 0$) exhibits error that increases with chain length. All proofs are deferred to Appendix A.2.

### A.2 Proofs

**Proof of Theorem A.1**

*Proof.* By the conditional bias–variance decomposition (law of total variance),

$$\mathbb{E}[\epsilon^2] = \mathbb{E}\big[\mathrm{Var}(\epsilon \mid x)\big] + \mathbb{E}\big[(\mathbb{E}[\epsilon \mid x])^2\big] = \mathbb{E}\big[\mathrm{Var}(\bar{\xi} \mid x)\big] + \mathbb{E}\big[\bar{m}^2\big].$$

The assumption $\mathrm{Var}(\bar{\xi} \mid x) \leq \tau^2_{\mathrm{orm}}$ for all $x$ gives $\mathbb{E}[\mathrm{Var}(\bar{\xi} \mid x)] \leq \tau^2_{\mathrm{orm}}$, and by definition $\beta^2_{\mathrm{orm}} = \mathbb{E}[\bar{m}^2]$. $\square$

**Proof of Theorem A.2.**

*Proof.* Let

$$B := \sum_{t=1}^{T} m_t, \qquad N := \sum_{t=1}^{T} \xi_t,$$

so $\Delta_{\mathsf{dPRM}} = B + N$ with $\mathbb{E}[N \mid x] = 0$. By the tower property,

$$\mathbb{E}\big[\Delta^2_{\mathsf{dPRM}}\big] = \mathbb{E}\big[\mathbb{E}[(B+N)^2 \mid x]\big] = \mathbb{E}\big[B^2 + 2B\mathbb{E}[N \mid x] + \mathbb{E}[N^2 \mid x]\big]$$
$$= \mathbb{E}\big[\mathrm{Var}(N \mid x)\big] + \mathbb{E}[B^2] \ \geq \ \mathbb{E}\big[\mathrm{Var}(N \mid x)\big].$$

Expanding the variance yields:

$$\mathrm{Var}(N \mid x) = \sum_{t=1}^{T} \mathrm{Var}(\xi_t \mid x) + 2 \sum_{1 \leq s < t \leq T} \mathrm{Cov}(\xi_s, \xi_t \mid x).$$

By local dependence, $\mathrm{Cov}(\xi_s, \xi_t \mid x) = 0$ when $|s - t| > k$. Hence only pairs with $t - s \leq k$ contribute. For each $t$, there are at most $k$ indices $s < t$ with $t - s \leq k$, so the number of contributing pairs is at most $kT$. Since each such covariance is bounded below by $-\rho$, we have

$$\sum_{1 \leq s < t \leq T} \mathrm{Cov}(\xi_s, \xi_t \mid x) \geq -k\rho T,$$

and therefore $2\sum_{s<t} \mathrm{Cov}(\cdot) \geq -2k\rho T$. Combining the variance floor and the covariance bound, for every fixed $x$,

$$\mathrm{Var}(N \mid x) = \sum_{t=1}^{T} \mathrm{Var}(\xi_t \mid x) + 2 \sum_{1 \leq s < t \leq T} \mathrm{Cov}(\xi_s, \xi_t \mid x)$$
$$\geq \sum_{t=1}^{T} \mathrm{Var}(\xi_t \mid x) - 2k\rho T$$
$$\geq T\sigma^2 - 2k\rho T.$$

Taking expectations over $x$ preserves inequalities, so

$$\mathbb{E}\big[\mathrm{Var}(N \mid x)\big] \geq \mathbb{E}\big[T\sigma^2 - 2k\rho T\big] = (\sigma^2 - 2k\rho)T.$$

Substituting back into the earlier bound $\mathbb{E}[\Delta_{\mathsf{dPRM}}^2] \geq \mathbb{E}[\mathrm{Var}(N \mid x)]$ yields

$$\mathbb{E}\big[\Delta_{\mathsf{dPRM}}^2\big] \geq (\sigma^2 - 2k\rho)T,$$

as claimed. $\qquad\square$

**Proof of Theorem A.3**

*Proof.* Decompose

$$\Delta_{\mathsf{gPRM}} = \sum_{t=1}^{T} \delta_t^{(g)} = \underbrace{\sum_{t=1}^{T} m_t^{(g)}}_{=:\, B^{(g)}} + \underbrace{\sum_{t=1}^{T} \xi_t^{(g)}}_{=:\, N^{(g)}}.$$

Conditional mean-zero $\mathbb{E}[N^{(g)} \mid x] = 0$ implies

$$\mathbb{E}\big[(\Delta_{\mathsf{gPRM}})^2\big] = \mathbb{E}\Big[\mathrm{Var}\Big(N^{(g)} \mid x\Big)\Big] + \mathbb{E}\Big[(B^{(g)})^2\Big] \geq \mathbb{E}\Big[\mathrm{Var}\Big(N^{(g)} \mid x\Big)\Big].$$

Now expand $\mathrm{Var}(N^{(g)} \mid x)$ using the average variance floor and local error dependence:

$$\mathrm{Var}(N^{(g)} \mid x) = \sum_{t=1}^{T} \mathrm{Var}(\xi_t^{(g)} \mid x) + 2 \sum_{1 \leq s < t \leq T} \mathrm{Cov}(\xi_s^{(g)}, \xi_t^{(g)} \mid x)$$
$$\geq T(\sigma^2 + \tau^2) - 2k\rho T.$$

Taking expectations in $x$ gives the stated bound. $\qquad\square$

**Proof of Theorem A.4**

*Proof.* **1) Exponential tilting and log-mgf.** Define $M(\theta) := \mathbb{E}[e^{\theta L} \mid x]$ and $K_x(\theta) := \log M(\theta)$. Since $e^{\theta L} \in (0,1]$ for $\theta \in [0,1]$ and $\mathbb{E}[|L|^2] < \infty$, dominated convergence yields $M'(\theta) = \mathbb{E}[Le^{\theta L} \mid x]$ and $M''(\theta) = \mathbb{E}[L^2 e^{\theta L} \mid x]$. Let $d\mathbb{P}_\theta(C) := e^{\theta L(x,C)} M(\theta)^{-1} d\mathbb{P}(C)$ and $\mathbb{E}_\theta[\cdot] := \mathbb{E}[\cdot\, e^{\theta L}]/M(\theta)$. Then

$$K'_x(\theta) = \frac{M'(\theta)}{M(\theta)} = \mathbb{E}_\theta[L \mid x], \qquad K''_x(\theta) = \frac{M''(\theta)M(\theta) - (M'(\theta))^2}{M(\theta)^2} = \mathrm{Var}_\theta(L \mid x).$$

**2) Jensen-gap identity.** Taylor with integral remainder at $\theta = 0$ gives

$$K_x(1) = K_x(0) + K'_x(0) + \int_0^1 (1 - \theta) K''_x(\theta)\, d\theta.$$

Since $K_x(0) = 0$ and $K'_x(0) = \mathbb{E}[L \mid x]$, we obtain

$$\log \mu(x) = \mathbb{E}[L \mid x] + \int_0^1 (1 - \theta)\, \mathrm{Var}_\theta(L \mid x)\, d\theta.$$

By definition of the mean predictor,

$$\Delta_{\mathrm{mean}}(x) = \log \mu(x) - \zeta(x), \text{ where } \mu(x) = \mathbb{E}[e^L \mid x].$$

Plugging $\log \mu(x) = \Delta_{\mathrm{mean}}(x) + \zeta(x)$ with $B^{(g)}(x) := \mathbb{E}[L \mid x] - \zeta(x)$, this yields

$$\Delta_{\mathrm{mean}}(x) = B^{(g)}(x) + \delta_J(x), \quad \delta_J(x) := \int_0^1 (1 - \theta)\, \mathrm{Var}_\theta(L \mid x)\, d\theta \geq 0.$$

**3) Lower bound on $\delta_J$ and variance linkage.** By tilt-stability,

$$\delta_J(x) \geq \frac{\kappa}{2} \operatorname{Var}(L \mid x).$$

Moreover, since $L = \zeta(x) + \Delta_{\mathsf{gPRM}} = \zeta(x) + B^{(g)} + N^{(g)}$ with $\mathbb{E}[N^{(g)} \mid x] = 0$, and since $\zeta(x)$ and $B^{(g)}(x)$ are constants when conditioning on $x$, we have

$$\operatorname{Var}(L \mid x) = \operatorname{Var}(N^{(g)} \mid x).$$

Expanding and using the average variance floors and local error dependence conditions (as in Theorem A.3),

$$\operatorname{Var}(N^{(g)} \mid x) \geq T(\sigma^2 + \tau^2) - 2k\rho T.$$

Combining this gives the pointwise bound

$$\Delta_{\mathrm{mean}}(x) \geq B^{(g)}(x) + \frac{\kappa}{2}\Big(T(\sigma^2 + \tau^2) - 2k\rho T\Big).$$

**4) Expectations and MSE.** Taking expectations over $x$ and applying Cauchy–Schwarz to $\mathbb{E}[B^{(g)}(x)]$ yields

$$\mathbb{E}[\Delta_{\mathrm{mean}}] \geq \frac{\kappa}{2}\Big((\sigma^2 + \tau^2 - 2k\rho)T\Big) - \sqrt{\mathbb{E}[B^{(g)}(x)^2]}.$$

Finally, Jensen's inequality gives $(\max\{0, \mathbb{E}[\Delta_{\mathrm{mean}}]\})^2 \leq \mathbb{E}[\Delta_{\mathrm{mean}}^2]$, so the MSE bound follows. In the calibrated case $B^{(g)} \equiv 0$, the stated simplified bounds hold. $\square$

# B   Prompts

In this section, we present prompt formats used in this work:

- Fig. 16: **User prompt format for generating CoTs** on GSM8K (Cobbe et al., 2021) and MATH (Hendrycks et al., 2021).
- Fig. 17: **User prompt format for generating CoTs** on MMLU-Pro (Wang et al., 2024c) proposed by Zeng et al. (2025).
- Fig. 18: **System prompt format for auto-labeling process labels** on MMLU-Pro (Wang et al., 2024c) proposed by Zeng et al. (2025).
- Fig. 19: **User prompt format for auto-labeling process labels** on MMLU-Pro (Wang et al., 2024c) proposed by Zeng et al. (2025).
- Fig. 20: **Prompt format of gORM** (Zhang et al., 2025a). We use this format for both generating synthetic verification-CoTs and training/evaluation of gORM.
- Fig. 21: **Prompt format for generating verification-CoTs** for gPRM following Khalifa et al. (2025).
- Fig. 22: **Prompt format of gPRM** for training and evaluation.

[**user**] Solve the following math problem efficiently and clearly:

- For simple problems (2 steps or fewer):
Provide a concise solution with minimal explanation.

- For complex problems (3 steps or more):
Use this step-by-step format:
## Step 1: [Concise description]
[Brief explanation and calculations]
## Step 2: [Concise description]
[Brief explanation and calculations]
`[OMITTED...]`

Regardless of the approach, always conclude with:
`Therefore, the final answer is: $\\boxed{answer}$.`

I hope it is correct. Where `[answer]` is just the final number or expression that solves the problem.

[Problem]
*{problem}*
[**/user**] [**assistant**]

Figure 16: **User prompt format for generating CoTs** on GSM8K (Cobbe et al., 2021) and MATH (Hendrycks et al., 2021).

[**user**] Given the following question and candidate answers, choose the best answer.
[Question]
*{question #1}*
[**/user**] [**assistant**]
*{assistant response #1}*
[**/assistant**]

[**user**] Given the following question and candidate answers, choose the best answer.
[Question]
*{question #2}*
[**/user**] [**assistant**]
*{assistant response #2}*
[**/assistant**]
`[OMITTED...]`

[**user**] Given the following question and candidate answers, choose the best answer.
[Question]
*{question}*
[**/user**] [**assistant**]

Figure 17: **User prompt format for generating CoTs** on MMLU-Pro (Wang et al., 2024c) proposed by Zeng et al. (2025)

[**system**] You are an experienced evaluator specializing in assessing the quality of reasoning steps in problem-solving. Your task is to find the first BAD step in a student's solution to a multiple choice question.

You will judge steps as GOOD, OK, or BAD based on the following criteria:

**1. GOOD Step** A step is classified as GOOD if it meets all of these criteria:

- **Correct**: Everything stated is accurate and aligns with known principles or the given problem.
- **Verifiable**: The step can be verified using common knowledge, simple calculations, or a quick reference (e.g., recalling a basic theorem). If verifying requires extensive effort (e.g., detailed calculations or obscure references), mark it BAD instead.
- **Appropriate**: The step fits logically within the context of the preceding steps. If a prior mistake exists, a GOOD step can correct it.
- **Insightful**: The step demonstrates reasonable problem-solving direction. Even if ultimately progressing in the wrong direction, it is acceptable as long as it represents a logical approach.

**2. OK Step** A step is classified as OK if it is:

- **Correct and Verifiable**: Contains no errors and can be verified.
- **Unnecessary or Redundant**: Adds little value, such as restating prior information or providing basic encouragement (e.g., "Good job!").
- **Partially Progressing**: Makes some progress toward the solution but lacks decisive or significant advancement.

**3. BAD Step** A step is classified as BAD if it:

- **Is Incorrect**: Contains factual errors, misapplies concepts, derives an incorrect result, or contradicts the ground truth answer.
- **Is Hard to Verify**: Requires significant effort to confirm due to poor explanation.
- **Is Off-Topic**: Includes irrelevant or nonsensical information.
- **Derails**: Leads to dead ends, circular reasoning, or unreasonable approaches.

**Task Description**

You will be provided with:

1. A Multiple Choice Question
2. A Ground Truth Answer
3. A Student's Step-by-Step Solution, where each step is enclosed with tags and indexed from 0.

Once you identify a BAD step, return the index of the earliest BAD step. Otherwise, return the index of -1 (which denotes all steps are GOOD or OK). Please put your final answer (i.e., the index) in \boxed{}.
[**/system**]

Figure 18: **System prompt format for auto-labeling process labels** on MMLU-Pro (Wang et al., 2024c) proposed by Zeng et al. (2025)

[**user**] The following is a multiple choice question and its ground truth answer. You are also given a student's solution (split into steps, enclosed with tags and indexed from 0):

[Multiple Choice Question]
*{question}*

[Ground Truth Answer]
*{answer}*

[Student Solution]
*{solution}*

[**/user**] [**assistant**] The first BAD step index is:

Figure 19: **User prompt format for auto-labeling process labels** on MMLU-Pro (Wang et al., 2024c) proposed by Zeng et al. (2025)

[**user**] You are a {`category`} teacher. Grade the solution, verifying correctness step by step.
At the end of Solution verification, when you give your final grade, write it in the form "Verification: Is the answer correct (Yes/No)? X", where X is either Yes or No.

[{`Category`} Problem]
{`problem`}

[Solution]
{`solution`}
[**/user**] [**assistant**] [**think**] Let's verify step by step:

Figure 20: **Prompt format of gORM** (Zhang et al., 2025a). We use this format for both generating synthetic verification-CoTs and training/evaluation of gORM.

[**user**] You are given a *{category}* problem and a proposed multiple-step solution (with a step on each line):

[*{Category}* Problem]
*{question}*

[Solution]
*{solution}*

Review and critique the proposed solution steps and determine whether each step is correct. If the solution is incomplete, only critique the steps that are provided. Your output must be in the following format:

Step 1: The step is \boxed{correct/incorrect}
Step 2: The step is \boxed{correct/incorrect}
$\vdots$
Step $n$: The step is \boxed{correct/incorrect}

Once you find an incorrect step, you should stop since you do not need to analyze the remaining steps. If the solution is incomplete, only verify the provided steps. [**/user**] [**assistant**] [**think**] Let's verify step by step:

Figure 21: **Prompt format for generating verification-CoTs** for gPRM following Khalifa et al. (2025).

> [**user**] You are given a {`category`} problem and a proposed step-by-step solution:
>
> [{`category`} Problem]
> {`problem`}
>
> [Solution]
> {`solution`}
>
> Review and critique each step in the proposed solution to determine whether each step is correct. If the solution is incomplete, only verify the provided steps. [**/user**] [**assistant**] [**think**] Let's verify step by step:

Figure 22: **Prompt format of gPRM** for training and evaluation.

## C   Dataset

In this section, we provide more details on the datasets.

**Math Datasets.**   For the math domain, we use the widely adopted **PRM800K** (Lightman et al., 2024) for training, where the process labels $z_{1:T}$ are human-annotated. For training ORMs, we set the outcome label $y = \mathbb{1}(z_{1:T} = \mathbf{1}_T)$ (rather than $y = \mathbb{1}(\hat{a}(r_T) = a)$), since PRM800K provides high-quality ground-truth process labels. As a testbed, we use **ProcessBench** (Zheng et al., 2024), which comprises four splits: 400 CoTs from GSM8K (Cobbe et al., 2021), 1K from Math (Hendrycks et al., 2021), 1K from Omni-Math (Gao et al., 2025), and 1K from OlympiadBench (He et al., 2024). We evaluate outcome verification by predicting $y \in \{0,1\}$ using the `final_answer_correct` field. We also generate $N = 16$ CoTs per question with Qwen2.5-7B-Instruct (Team, 2024a) to assess test-time scaling (TTS).

Table 1: Dataset statistics for each domain of MMLU-pro (Wang et al., 2024c). We report the number of questions, the number of CoTs, and the average number of CoTs per question for both training and test splits.

| Domain | Training Set | | | Test Set | | |
|---|---|---|---|---|---|---|
| | # Questions | # CoTs | Avg. CoTs / Q | # Questions | # CoTs | Avg. CoTs / Q |
| Law | 500 | 7,806 | 15.61 | 145 | 18,537 | 127.84 |
| Psychology | 498 | 7,901 | 15.87 | 150 | 19,164 | 127.76 |
| Chemistry | 500 | 6,537 | 13.07 | 150 | 15,981 | 106.54 |
| Biology | 417 | 6,420 | 15.40 | 130 | 16,441 | 126.47 |
| Physics | 500 | 6,680 | 13.36 | 150 | 16,460 | 109.73 |
| History | 81 | 1,275 | 15.74 | 150 | 19,159 | 127.73 |
| Economics | 500 | 7,749 | 15.50 | 150 | 18,911 | 126.07 |
| Math | 500 | 6,940 | 13.88 | 150 | 17,014 | 113.43 |
| Business | 489 | 6,969 | 14.25 | 149 | 17,344 | 116.40 |
| Philosophy | 199 | 3,125 | 15.70 | 149 | 18,844 | 126.47 |
| Health | 456 | 7,202 | 15.79 | 140 | 17,862 | 127.59 |
| Engineering | 500 | 6,032 | 12.06 | 150 | 15,708 | 104.72 |
| Computer Science | 110 | 1,638 | 14.89 | 150 | 18,429 | 122.86 |
| Other | 500 | 7,824 | 15.65 | 150 | 18,982 | 126.55 |
| **Total** | 5,750 | 84,098 | 14.63 | 2,063 | 248,836 | 120.62 |

**Multi-domain datasets.**   For the multi-domain setting, we adopt **MMLU-Pro** (Wang et al., 2024c), a 10-choice benchmark spanning 14 domains: law, psychology, chemistry, biology, physics, history, economics, math, business, philosophy, health, engineering, computer science, and other. As shown in Table 1, the corpus includes 5,750 training and 2,063 evaluation questions. For each question, Zeng et al. (2025) generate 16/128 CoTs for training/evaluation with Llama-3.1-8B-Instruct (Dubey et al., 2024), and auto-label reasoning steps (*i.e.*, process labels) using Llama-3.1-70B-Instruct with prompts in Figs. 18 and 19; please see Zeng et al. (2025) for more details. To assess generalization across CoTs from different $p_{\text{LLM}}$, we also generate 16 CoTs per evaluation question using SmolLM3-3B (Bakouch et al., 2025), Qwen2.5-7B-Instruct, gemma-2-9b-it (Team et al., 2024), and Llama-3.1-70B-Instruct, spanning diverse model sizes and families.

## D  Implementation Details

In this section, we provide implementation details omitted from the main paper due to space limits.

**Backbones for reward models.**  Following Zhang et al. (2025a) and Khalifa et al. (2025), we use R1-Distill-Qwen-1.5B and R1-Distill-Qwen-7B (Guo et al., 2025) for the math domain, and R1-Distill-Llama-8B and R1-Distill-Qwen-14B for the multi-domain setting, as reward-model backbones. We also use Qwen3-8B as the backbones to assess whether the results hold for non-distilled backbones in the multi-domain setting. Note that VersaPRM (Zeng et al., 2025) originally used Llama-3.1-8B-Instruct as the reward-model backbone for dPRM; for a fair comparison, we use R1-Distill models and Qwen3-8B for both dORM and dPRM.

Table 2: **Summary of hyperparameters**.

| Method | LoRA | | | Training | | | | | | Inference | | |
|---|---|---|---|---|---|---|---|---|---|---|---|---|
| | Rank $r$ | $\alpha$ | Dropout $p$ | Batch | Optim. | Epochs | LR | Decay | Scheduler | Package | Temp. $\tau$ | $M$ |
| dORM & dPRM | 16 | 32 | 0.1 | 16 | AdamW | 1 | 1e-4 | 1e-2 | Cosine | - | - | - |
| gORM & gPRM | 32 | 16 | 0.1 | 16 | AdamW | 1 | 1e-4 | 1e-2 | Linear | vLLM | 0.6 | 10 or 16 |

**Hyperparameters.**  We apply LoRA (Hu et al., 2022) for parameter-efficient fine-tuning, optimize with AdamW (Loshchilov & Hutter, 2019), and use vLLM (Kwon et al., 2023) for fast inference. At inference, we sample $M=16$ verification CoTs for the math domain and $M=10$ for the multi-domain setting. Hyperparameters are summarized in Table 2: for dORM/dPRM we adopt those of Zeng et al. (2025), and for gORM/gPRM we follow Khalifa et al. (2025). Note that in preliminary experiments we set $r=32$ and $\alpha=16$ for dORM/dPRM to compare fairly with gORM/gPRM (also using $r=32$ and $\alpha=16$). However, we observed an **overall performance degradation** (*e.g.*, $\approx 2\%$), so we follow the settings of Zeng et al. (2025). The hyperparameters in Table 2 are shared across all experiments and we do not perform exhaustive tuning[4]. We report means over three independent runs for all experiments.

**Verification CoTs for gORM and gPRM.**  Following Khalifa et al. (2025), we sample 4 different verification CoTs for each question $q$ and CoT $r_{1:T}$ pair in the training dataset by prompting QwQ-32B (Qwen Team, 2025) with `temperature=0.6`, `top_k=20`, `top_p=0.95`, and `min_p=0` using the formats in Figs. 20 and 21. Note that Khalifa et al. (2025) originally used QwQ-32B-Preview (Team, 2024b). In preliminary experiments, we found QwQ-32B more likely to follow instructions and produce more parsable verification CoTs (*e.g.*, 1K vs. 7K for gPRM in the law domain), so we use QwQ-32B throughout.

For the math domain we set `category` as `math`; for the multi-domain setting we use `category` $\in \{$law, ..., `computer science`$\}$ and leave it blank for `other`. For additional benchmarks, such as GPQA-diamond, MedQA, and LEXam, we also leave `category` blank. We discard any verification CoT that (i) has **unparsable labels**; (ii) contains **Chinese characters**; (iii) exceeds **the token limit**—4,096 for math (Khalifa et al., 2025) or 8,192 for multi-domain; or (iv) whose parsed labels are **inconsistent with the targets** (*e.g.*, $y$ or $z_{1:T}$), corresponding to the *consensus filtering* in §2.2. We also balance the counts of `Yes`/`No` examples. The resulting training sets contain 34,286 CoTs for gORM and 35,666 CoTs for gPRM in the math domain, and 171,780 (gORM) and 94,156 (gPRM) in the multi-domain setting.

## E  Training Examples

In this section, we present training examples including verification CoTs:

- Fig. 23: **Training example of gORM** on the law domain of MMLU-pro (Wang et al., 2024c).
- Fig. 24: **Training example of gPRM** on the law domain of MMLU-pro (Wang et al., 2024c).

---

[4]In Table 3, we report the results of sweeps over the learning rate and LoRA rank for dPRM/gPRM.

---

[user]
You are a {category} teacher. Grade the solution, verifying correctness step by step.
At the end of Solution verification, when you give your final grade, write it in the form "Verification: Is the answer correct (Yes/No)? X", where X is either Yes or No.

[Law Problem]
Question: A shopper in a store sees that a display case is about to fall on a second shopper's head. The shopper does nothing to warn or assist the second shopper, who is a stranger to her. She could have taken action to protect the second shopper without harming herself. Did the shopper have a duty to take action to try and protect the second shopper from injury?
A. No, the shopper is not required to intervene unless she is a store employee or security personnel.
B. Yes, the shopper must take action under the G̈ood Samaritanl̈aw.
C. Yes, the fact that a person sees that action is needed for another's aid or protection does impose a legal duty to act. [OMITTED...]

[Solution]

The shopper could have taken action to protect the second shopper without harming herself, so the issue is whether there is a legal duty to intervene.

The "Good Samaritan" law protects individuals from liability for failing to provide assistance, so B is incorrect.

There is no special relationship between the shopper and the second shopper, so I is incorrect.

[OMITTED...]

Therefore, the shopper is not required to intervene, and the correct answer is D.

The answer is (D).
[/user] [assistant] [think]

Let's verify step by step:
The question is about whether a shopper has a duty to act to prevent harm to another stranger when she can do so without risk. The solution concludes D is correct.

First, the solution correctly dismisses B because Good Samaritan laws typically protect those who do help, not impose a duty to act. That seems right.

For option I, the solution says no special relationship exists. Since the shopper and victim are strangers, that's accurate. So I is wrong.

[OMITTED...]

The solution mentions "duty to rescue" in certain circumstances like special relationships or being first to notice. However, in the problem, there's no indication the shopper was first. So D holds.

Wait, but the problem states "she could have taken action without harming herself," which is a key point. Some jurisdictions might have a duty if it's safe, but traditionally, common law doesn't. The solution's conclusion aligns with that.

Therefore, the answer D is correct. The solution's reasoning is sound, dismissing other options properly. The final answer D matches the analysis. So the solution is correct.

[/think]

Verification: Is the answer correct (Yes/No)? Yes [/assistant]

Figure 23: **Training example of gORM** on the law domain of MMLU-Pro (Wang et al., 2024c).

---

[**user**] You are given a law problem and a proposed step-by-step solution:

[Law Problem]
Question: A plaintiff sued a defendant for injuries that the plaintiff suffered when he was struck by the defendant's car. At trial, the plaintiff testified that he was walking across the street inside a crosswalk at the time the defendant's car hit him. `[OMITTED...]` In support of the defendant's testimony, the defendant calls a neighbor to testify that she has observed the plaintiff frequently during the years that she has lived on that street, and that the plaintiff "always crosses the street outside the crosswalk." The neighbor's testimony is likely to be
A. admitted, because it tends to show that the plaintiff was careless.
`[OMITTED...]`
G. admitted, because it is probative of the plaintiff's habit of crossing the street outside the crosswalk.
H. excluded, because it is irrelevant to the specific incident.
I. admitted, because it provides context to the plaintiff's behavior.

[Solution]
Step 1: The neighbor's testimony is about the plaintiff's habit of crossing the street outside the crosswalk, which is relevant to the case.
Step 2: Habit evidence is admissible to prove that a person regularly acts in conformity with the habit.
Step 3: The neighbor's testimony is an example of habit evidence, as it describes the plaintiff's regular behavior of crossing the street outside the crosswalk.
Step 4: The answer is (G).

Review and critique each step in the proposed solution to determine whether each step is correct. If the solution is incomplete, only verify the provided steps. [**/user**] [**assistant**] [**think**]
Let's verify step by step:

First, the problem is about whether the neighbor's testimony about the plaintiff's habit is admissible. The proposed solution says step 1 is correct because the testimony is about habit and relevant. That seems right since the case hinges on where the plaintiff was crossing.

Step 2 states habit evidence is admissible to show conformity. Under Federal Rules of Evidence, habit (as opposed to character) is admissible. So step 2 is correct.

Step 3 says the neighbor's testimony is habit evidence. But wait, habit requires a regular, routine practice. The neighbor says "frequently" over years, which might qualify as a habit. So step 3 is correct.

Step 4 concludes G is correct. So all steps seem okay. Wait, but maybe "frequently" isn't enough? Some jurisdictions require more specific regularity. But generally, habit evidence can be admissible if it's a settled pattern. So steps are correct. Hmm, but let me check again. The options: G says "probative of plaintiff's habit". So yes, the steps are correct. The answer should be G. So all steps are correct.
[**/think**]

Step 1: The step is \boxed{correct}
Step 2: The step is \boxed{correct}
Step 3: The step is \boxed{correct}
Step 4: The step is \boxed{correct}
Is the solution correct? Yes
[**/assistant**]

Figure 24: **Training example of gPRM** on the law domain of MMLU-Pro (Wang et al., 2024c).

# F   Detailed Results on MMLU-Pro

In this section, we present detailed results on MMLU-Pro (Wang et al., 2024c) as follows:

- Fig. 25: **Best-of-$N$ results using Llama-3.1-8B-Instruct** on MMLU-Pro with R1-Distill-Qwen-14B backbone for reward models.

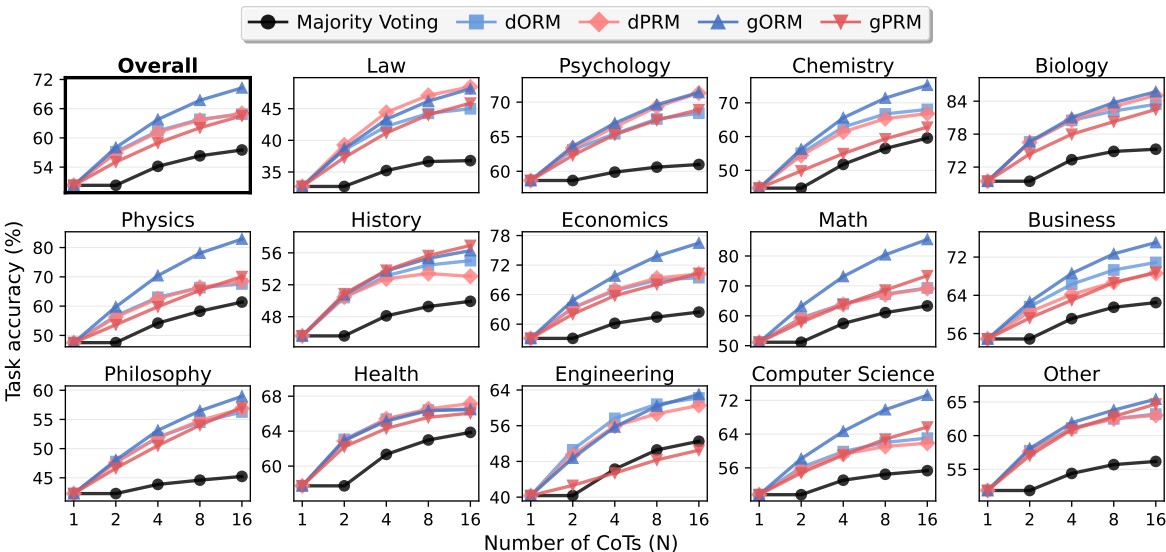

Figure 25: **Best-of-$N$ results using Llama-3.1-8B-Instruct** on MMLU-Pro with R1-Distill-Qwen-14B backbone for reward models.

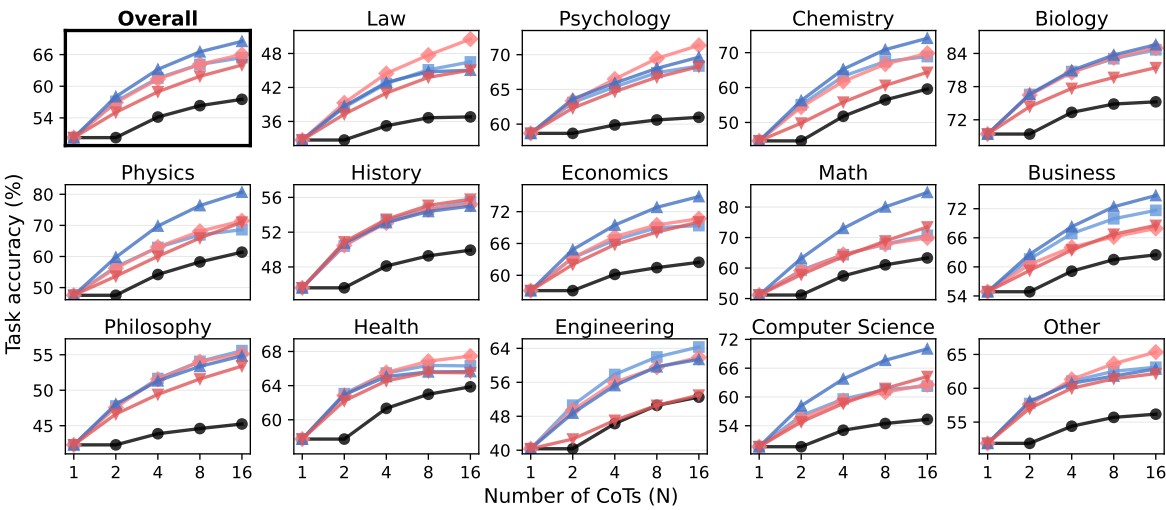

Figure 26: **Weighted majority voting results using Llama-3.1-8B-Instruct** on MMLU-Pro with R1-Distill-Qwen-14B backbone for reward models.

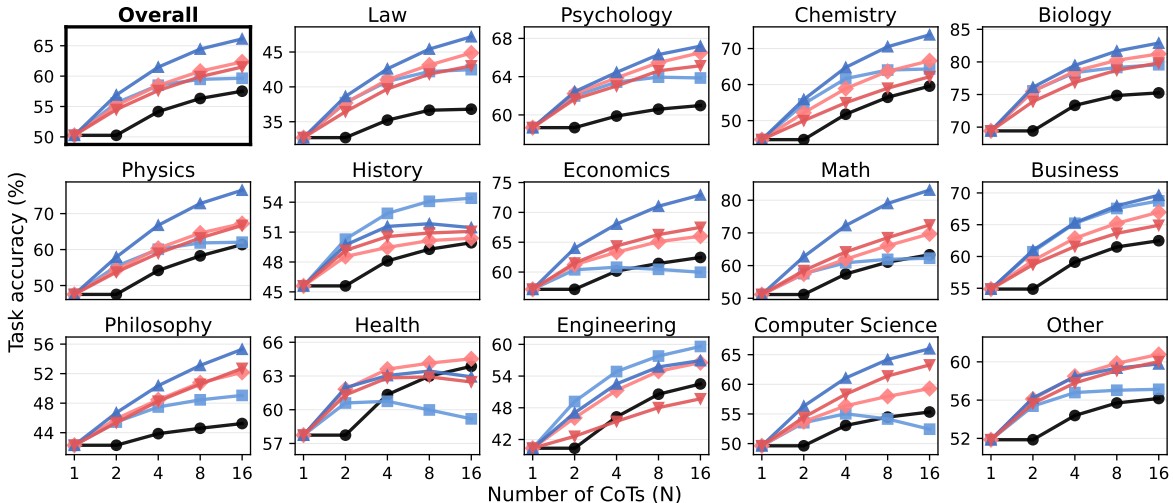

Figure 27: **Best-of-$N$ results using Llama-3.1-8B-Instruct** on MMLU-Pro (Wang et al., 2024c) with R1-distill-Llama-8B backbone for reward models.

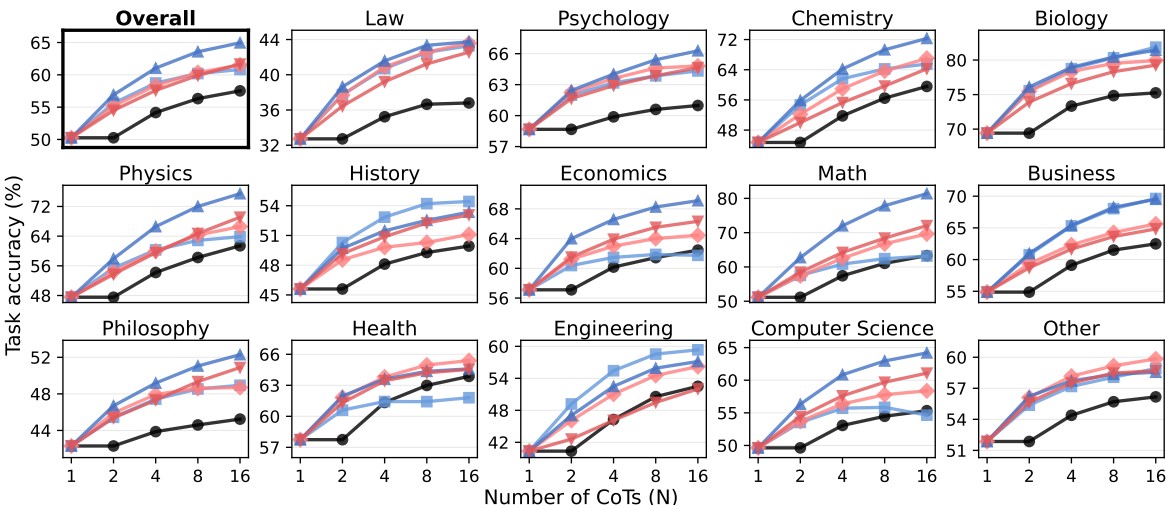

Figure 28: **Weighted majority voting results using Llama-3.1-8B-Instruct** on MMLU-Pro (Wang et al., 2024c) with R1-Distill-Llama-8B backbone for reward models.

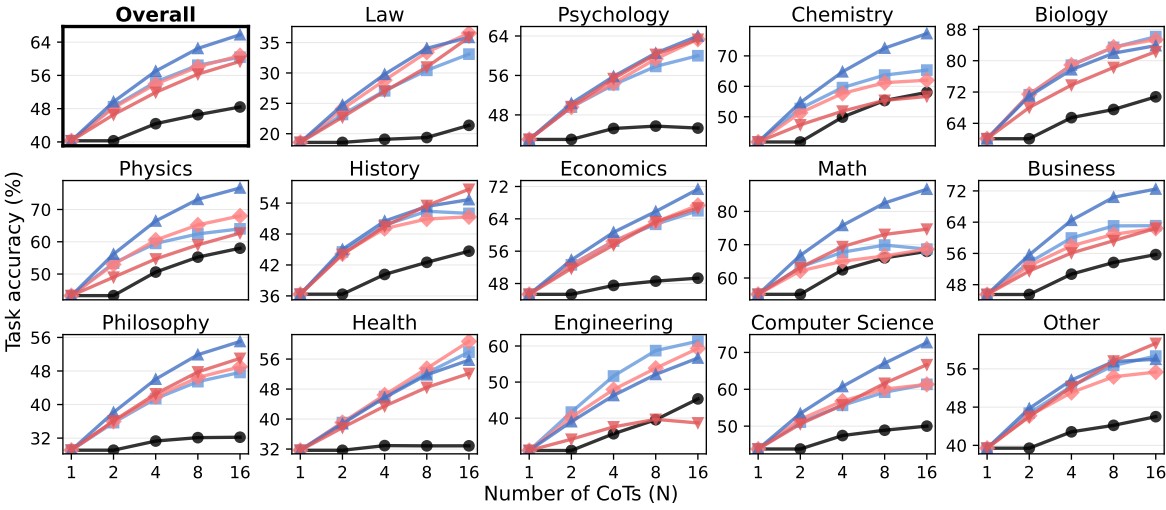

Figure 29: **Best-of-$N$ results using SmolLM3-3B** on MMLU-Pro with R1-Distill-Qwen-14B backbone for reward models.

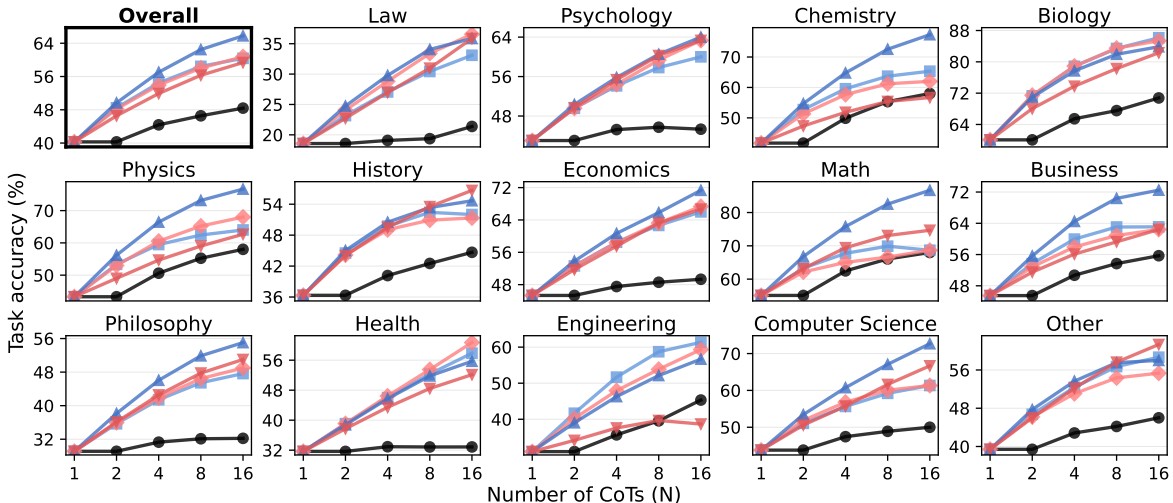

Figure 30: **Weighted majority voting results using SmolLM3-3B** on MMLU-Pro with R1-Distill-Qwen-14B backbone for reward models.

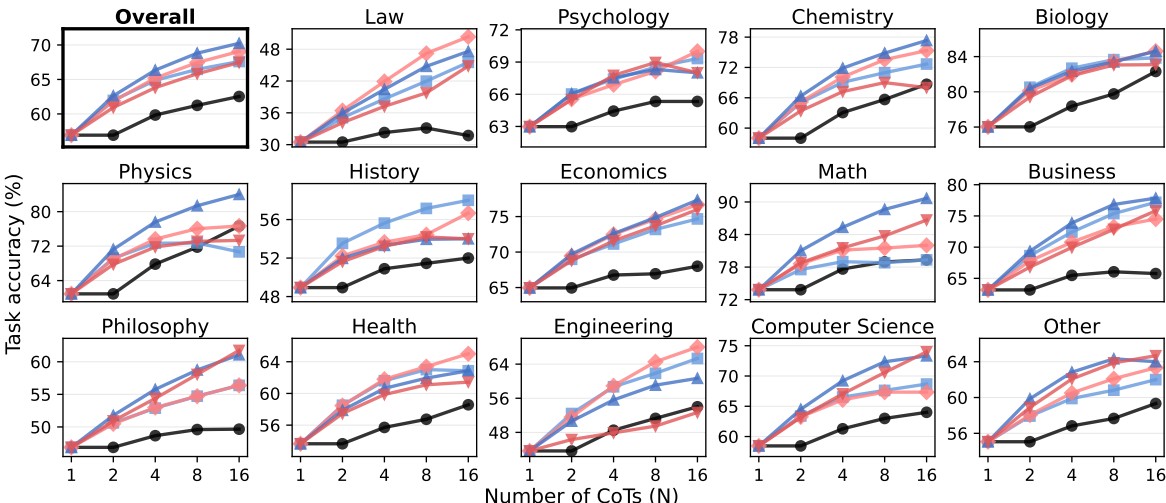

Figure 31: **Best-of-$N$ results using Qwen2.5-7B-Instruct** on MMLU-Pro with R1-Distill-Qwen-14B backbone for reward models.

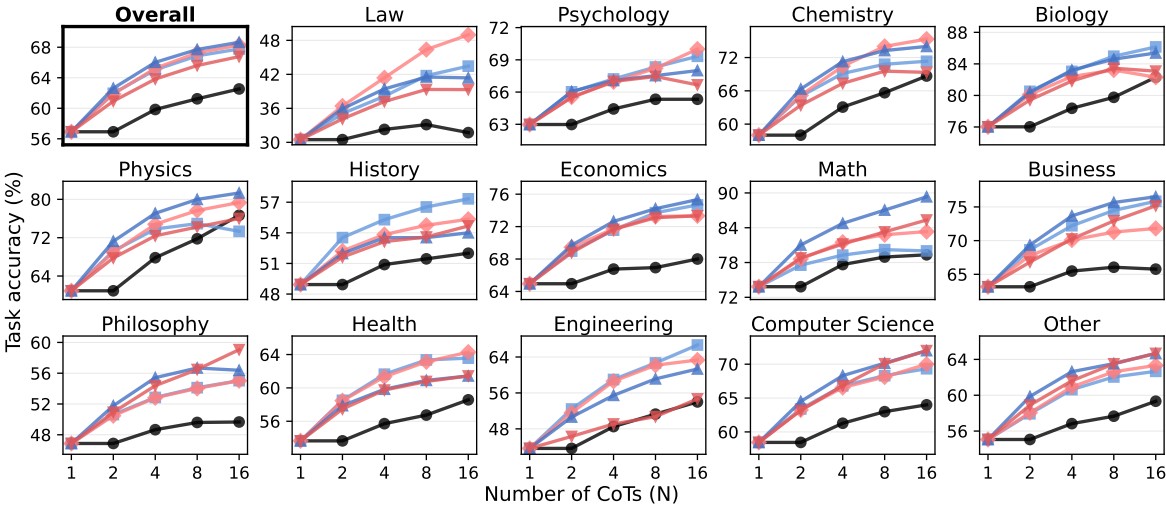

Figure 32: **Weighted majority voting results using Qwen2.5-7B-Instruct** on MMLU-Pro with R1-Distill-Qwen-14B backbone for reward models.

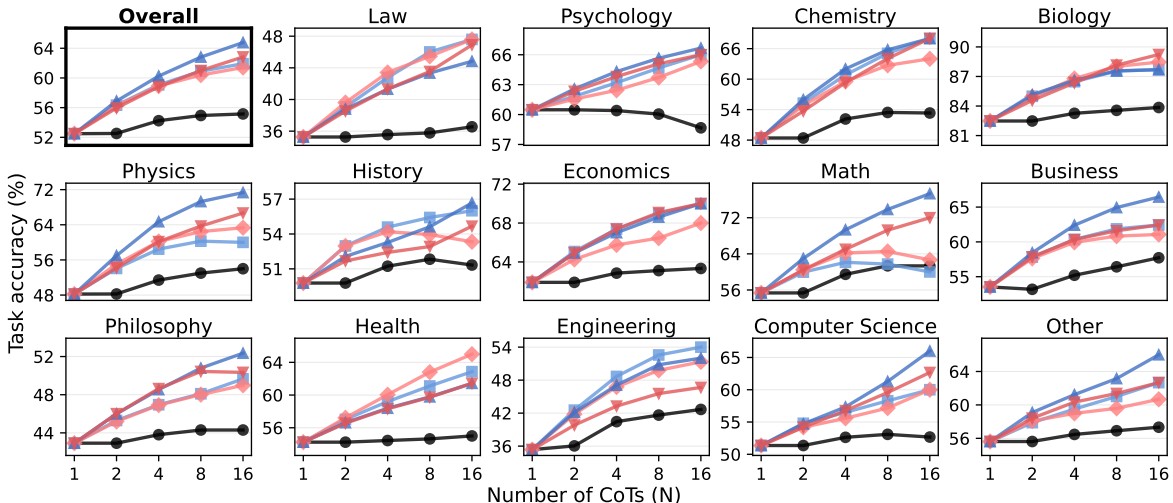

Figure 33: **Best-of-$N$ results using gemma-2-9b-it** on MMLU-Pro with R1-Distill-Qwen-14B backbone for reward models.

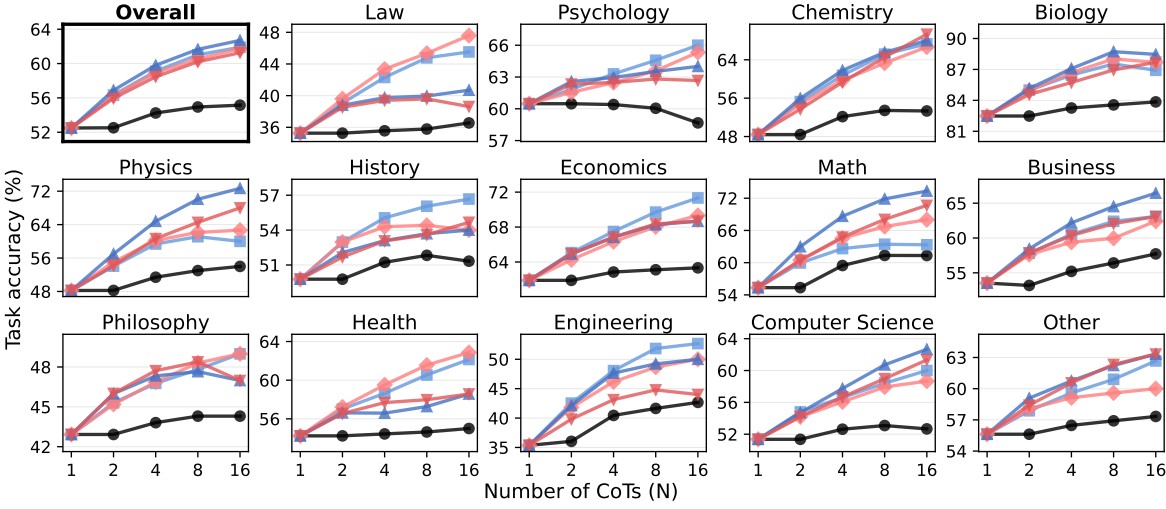

Figure 34: **Weighted majority voting results using gemma-2-9b-it** on MMLU-Pro with R1-Distill-Qwen-14B backbone for reward models.

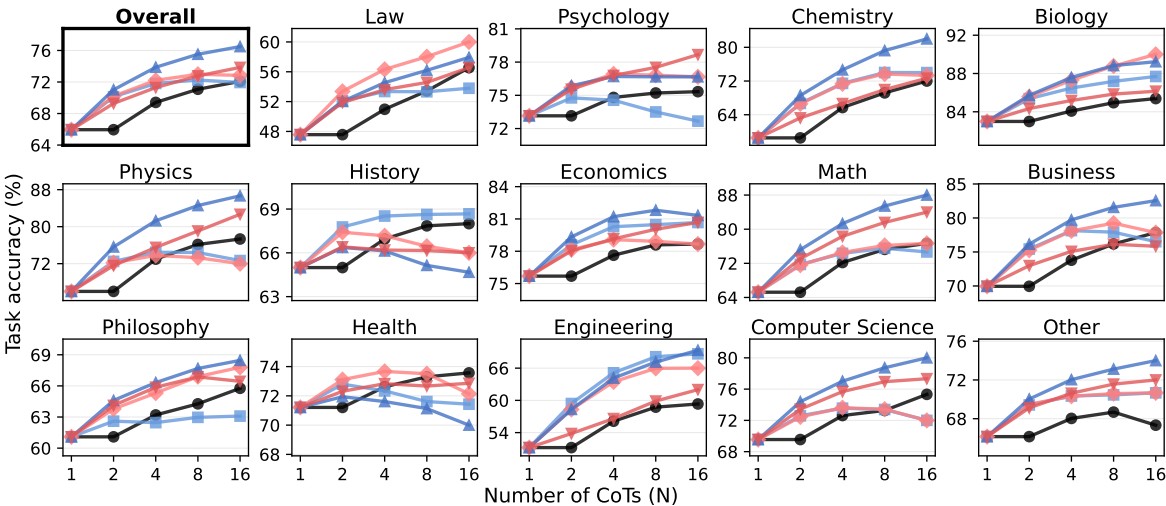

Figure 35: **Best-of-$N$ results using Llama-3.1-70B-Instruct** on MMLU-Pro with R1-Distill-Qwen-14B backbone for reward models.

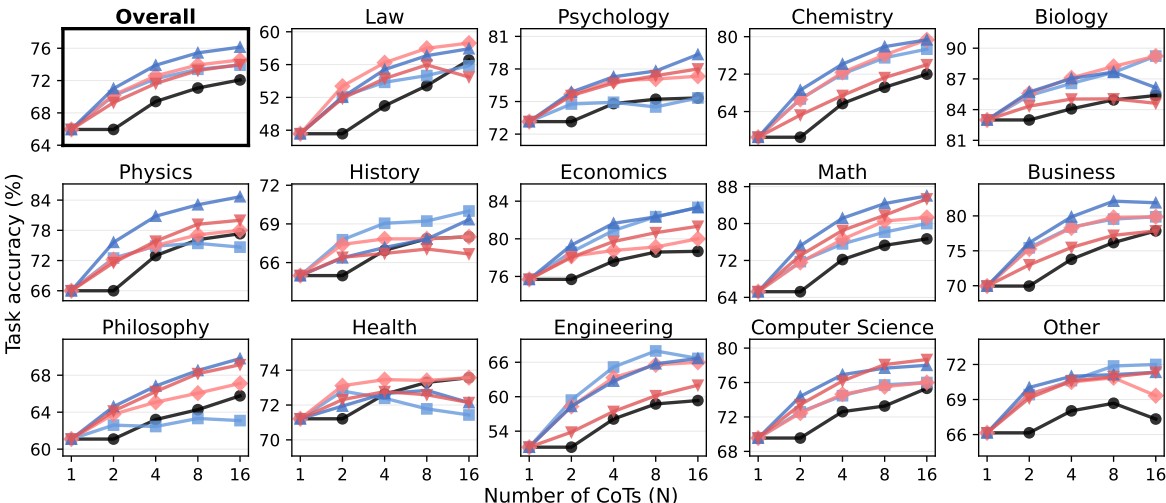

Figure 36: **Weighted majority voting results using Llama-3.1-70B-Instruct** on MMLU-Pro with R1-Distill-Qwen-14B backbone for reward models.

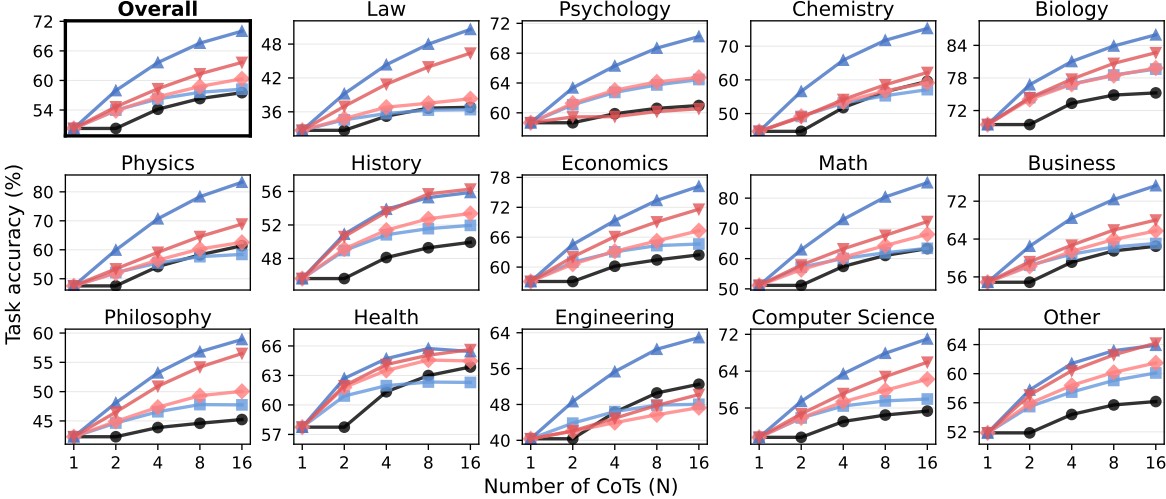

Figure 37: **Best-of-$N$ performance using Llama-3.1-8B-Instruct when trained and evaluated on each domain** of MMLU-Pro with R1-Distill-Qwen-14B backbone for reward models.

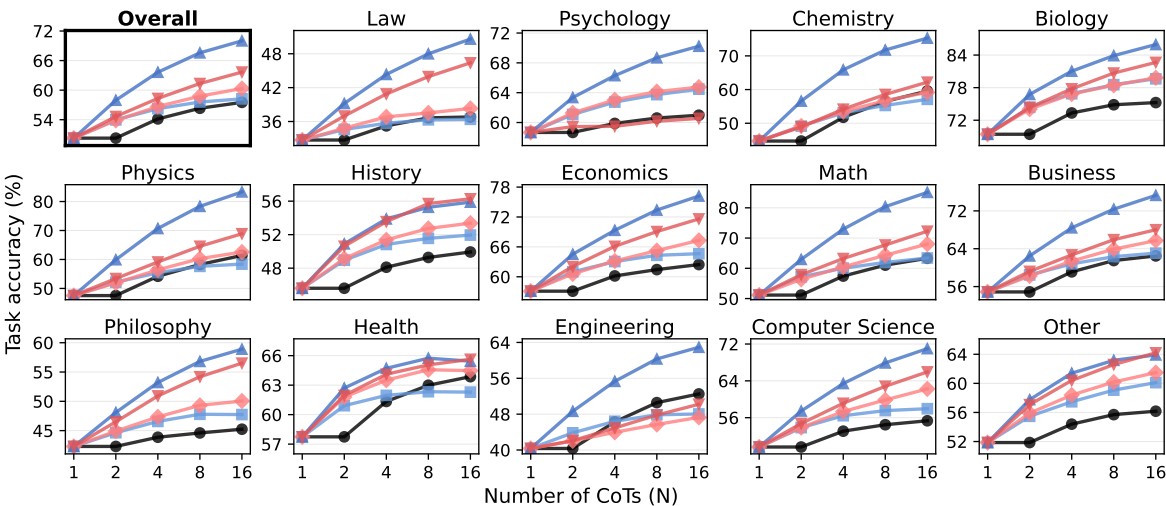

Figure 38: **Weighted majority voting performance using Llama-3.1-8B-Instruct when trained and evaluated on each domain** of MMLU-Pro with R1-Distill-Qwen-14B backbone for reward models.

# G Additional Analysis

In this section, we present additional analysis on the failure of PRMs.

- Fig. 39: Overall Best-of-$N$ results using five different $p_{\mathtt{LLM}}$ on MMLU-Pro with **Qwen3-8B** backbone for reward models.

- Table 3: Best-of-$N$ results (overall) on MMLU-Pro using Llama-3.1-8B-Instruct by **varying learning rate and LoRA rank $r$ for PRM variants**. We use R1-Distill-Qwen-14B backbone for reward models.

- Table 4: **Wasserstein distance in the math domain** before and after filtering for gORM and gPRM.

- Table 5: **Wasserstein distance in the multi-domain setting** before and after filtering for gORM and gPRM. To reduce the CoT-length distribution shift (*i.e.*, the Wasserstein distance) for gPRM, we apply (i) **label refinement** using Gemini-2.0 Flash (Comanici et al., 2025): due to a parsing issue, **59.96%** of process labels are replaced; and (ii) **relaxation** of the *consensus filtering* rule: when $y = 1$, we keep the verification CoTs $v_{1:L^+}$ with $\hat{z}_{1:T} = 1_T$, and when $y = 0$, we keep $v_{1:L^+}$ if there exists $t \in \{1, \ldots, T\}$ such that $z_t = 0$.

- Table 6: **Surviving proportion (%) of CoTs** on the train split of MMLU-Pro. We compare gORM and gPRM under (i) **label refinement** using Gemini-2.0 Flash (Comanici et al., 2025), and (ii) **relaxed** *consensus filtering*.

- Table 7: Best-of-$N$ results on MMLU-Pro using Llama-3.1-8B-Instruct with (i) **label refinement** or (ii) **relaxed filtering**. We use R1-Distill-Qwen-14B as the backbone for reward models.

- Table 8: Best-of-$N$ results on MMLU-Pro using CoTs generated by SmolLM3-3B. We use R1-Distill-Qwen-14B as the backbone for reward models.

- Table 9: Best-of-$N$ results on MMLU-Pro using CoTs generated by Qwen2.5-7B-Instruct. We use R1-Distill-Qwen-14B as the backbone for reward models.

- Table 10: Best-of-$N$ results on MMLU-Pro using CoTs generated by Llama-3.1-8B-Instruct. We use R1-Distill-Qwen-14B as the backbone for reward models.

- Table 11: Best-of-$N$ results on MMLU-Pro using CoTs generated by gemma2-9b-it. We use R1-Distill-Qwen-14B as the backbone for reward models.

- Table 12: Best-of-$N$ results on MMLU-Pro using CoTs generated by Llama-3.1-70B-Instruct. We use R1-Distill-Qwen-14B as the backbone for reward models.

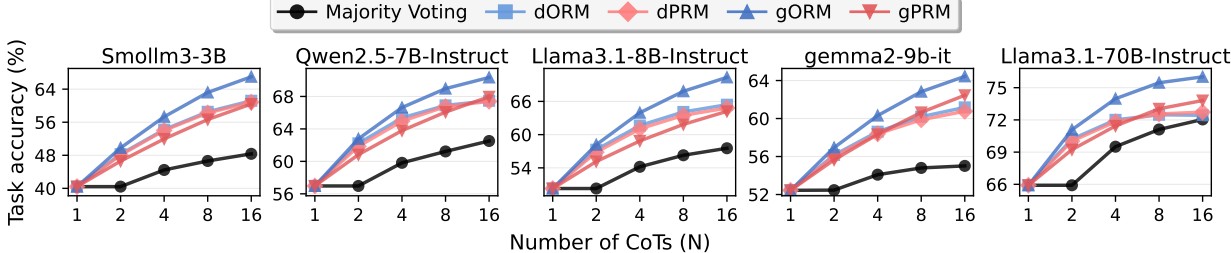

Figure 39: Overall Best-of-$N$ results using five different $p_{\text{LLM}}$ on MMLU-Pro with **Qwen3-8B** backbone for reward models.

Table 3: Best-of-$N$ results (overall) on MMLU-Pro using Llama-3.1-8B-Instruct by **varying learning rate and LoRA rank $r$ for PRM variants**. We use with R1-Distill-Qwen-14B backbone for reward models. The number in parentheses denotes the change.

| Method | Learning rate | $r$ | $N$ | | | | |
|---|---|---|---|---|---|---|---|
| | | | 1 | 2 | 4 | 8 | 16 |
| **Majority voting** | – | – | 50.27 | 50.27 | 54.15 | 56.14 | 57.16 |
| **dORM** | $1 \cdot 10^{-4}$ | 16 | 50.27 | 57.30 | 61.54 | 63.95 | 65.38 |
| **dPRM (default)** | $1 \cdot 10^{-4}$ | 16 | 50.27 | 57.18 | 61.51 | 64.10 | 65.55 |
| **dPRM (changed)** | $5 \cdot 10^{-5}$ | 16 | 50.27 | 56.77 **(-0.41)** | 60.84 **(-0.67)** | 62.99 **(-1.11)** | 64.04 **(-1.51)** |
| **dPRM (changed)** | $1 \cdot 10^{-4}$ | 32 | 50.27 | 57.01 **(-0.17)** | 61.31 **(-0.20)** | 64.11 **(+0.01)** | 66.02 **(+0.47)** |
| **gORM** | $1 \cdot 10^{-4}$ | 32 | 50.27 | 58.24 | 63.88 | 67.82 | 70.02 |
| **gPRM (default)** | $1 \cdot 10^{-4}$ | 32 | 50.27 | 55.24 | 59.06 | 62.10 | 64.26 |
| **gPRM (changed)** | $5 \cdot 10^{-5}$ | 32 | 50.27 | 55.09 **(-0.15)** | 58.94 **(-0.12)** | 61.96 **(-0.14)** | 64.26 **(+0.00)** |
| **gPRM (changed)** | $1 \cdot 10^{-4}$ | 64 | 50.27 | 54.94 **(-0.30)** | 58.73 **(-0.33)** | 61.88 **(-0.22)** | 64.55 **(+0.29)** |
| **Pass@$N$** | – | – | 50.27 | 61.74 | 71.56 | 79.77 | 86.05 |

Table 4: **Wasserstein distance in the math domain** before and after filtering for gORM and gPRM.

| | Overall | GSM8K | Math | Omni-Math | OlympiadBench |
|---|---|---|---|---|---|
| Train (PRM800K) | 2.760 | 5.113 | 3.813 | 2.027 | 1.514 |
| gORM | 2.430 | 4.780 | 3.480 | 1.695 | 1.194 |
| gPRM | 1.600 | 3.680 | 2.348 | 1.448 | 1.203 |

Table 5: **Wasserstein distance in the multi-domain setting** before and after filtering for gORM and gPRM. To reduce the CoT-length distribution shift (*i.e.*, the Wasserstein distance) for gPRM, we apply (i) **label refinement** using Gemini-2.0 Flash (Comanici et al., 2025): due to a parsing issue, **59.96%** of process labels are replaced; and (ii) **relaxation** of the *consensus filtering* rule: when $y = 1$, we keep the verification CoTs $v_{1:L^+}$ with $\hat{z}_{1:T} = 1_T$, and when $y = 0$, we keep $v_{1:L^+}$ if there exists $t \in \{1, \dots, T\}$ such that $z_t = 0$.

| | Overall | Law | Psychology | Chemistry | Biology |
|---|---|---|---|---|---|
| Train | 0.202 | 0.090 | 0.203 | 0.393 | 0.264 |
| gORM | 0.532 | 0.089 | 0.218 | 1.128 | 0.506 |
| gPRM | 3.083 | 1.284 | 0.742 | 6.922 | 2.039 |
| gPRM + label refinement | 3.265 **(+0.182)** | 1.397 **(+0.113)** | 0.893 **(+0.151)** | 7.183 **(+0.261)** | 2.416 **(+0.377)** |
| gPRM + relaxed filtering | 2.001 **(-1.082)** | 0.885 **(-0.399)** | 0.397 **(-0.345)** | 4.939 **(-1.983)** | 1.311 **(-0.728)** |
| | Physics | History | Economics | Math | Business |
| Train | 0.628 | 0.069 | 0.311 | 0.167 | 0.322 |
| gORM | 1.201 | 0.154 | 0.564 | 0.282 | 0.491 |
| gPRM | 5.952 | 0.581 | 1.782 | 4.655 | 4.267 |
| gPRM + label refinement | 6.104 **(+0.152)** | 0.752 **(+0.171)** | 2.044 **(+0.262)** | 4.852 **(+0.197)** | 4.494 **(+0.227)** |
| gPRM + relaxed filtering | 4.371 **(-1.581)** | 0.203 **(-0.378)** | 1.094 **(-0.688)** | 2.571 **(-2.084)** | 2.777 **(-1.490)** |
| | Philosophy | Health | Engineering | Computer science | Other |
| Train | 0.129 | 0.105 | 1.234 | 0.353 | 0.093 |
| gORM | 0.545 | 0.213 | 3.611 | 0.338 | 0.312 |
| gPRM | 1.235 | 0.979 | 12.735 | 3.459 | 0.927 |
| gPRM + label refinement | 1.299 **(+0.064)** | 1.157 **(+0.178)** | 13.058 **(+0.323)** | 3.742 **(+0.283)** | 1.030 **(+0.103)** |
| gPRM + relaxed filtering | 0.505 **(-0.730)** | 0.554 **(-0.425)** | 9.536 **(-3.199)** | 2.363 **(-1.096)** | 0.460 **(-0.467)** |

Table 6: **Surviving proportion (%) of CoTs** on the train split of MMLU-Pro. We compare gORM and gPRM under (i) **label refinement** using Gemini-2.0 Flash (Comanici et al., 2025), and (ii) **relaxed** *consensus filtering*. Please see the caption of Table 5 for more details.

|  | Overall | Law | Psychology | Chemistry | Biology |
|---|---|---|---|---|---|
| gORM | 51.1 | 51.6 | 28.3 | 71.9 | 42.0 |
| gPRM | 28.0 | 22.7 | 22.8 | 30.1 | 30.4 |
| gPRM + label refinement | 44.0 (**+16.0**) | 23.6 (**+0.9**) | 42.1 (**+19.3**) | 33.5 (**+3.4**) | 55.4 (**+25.0**) |
| gPRM + relaxed filtering | 45.5 (**+17.5**) | 24.9 (**+2.2**) | 43.7 (**+20.9**) | 35.8 (**+5.7**) | 54.6 (**+24.2**) |
|  | Physics | History | Economics | Math | Business |
| gORM | 77.0 | 28.4 | 50.9 | 70.3 | 54.3 |
| gPRM | 34.2 | 26.0 | 37.0 | 30.8 | 24.7 |
| gPRM + label refinement | 35.8 (**+1.6**) | 49.3 (**+23.3**) | 64.9 (**+27.9**) | 65.5 (**+34.7**) | 56.0 (**+31.3**) |
| gPRM + relaxed filtering | 37.3 (**+3.1**) | 41.0 (**+15.0**) | 64.1 (**+27.1**) | 66.7 (**+35.9**) | 57.8 (**+33.1**) |
|  | Philosophy | Health | Engineering | Computer Science | Other |
| gORM | 48.6 | 40.5 | 37.0 | 70.1 | 42.9 |
| gPRM | 26.8 | 27.7 | 13.5 | 38.9 | 31.0 |
| gPRM + label refinement | 46.6 (**+19.8**) | 51.3 (**+23.6**) | 13.9 (**+0.4**) | 46.9 (**+8.0**) | 37.5 (**+6.5**) |
| gPRM + relaxed filtering | 48.0 (**+21.2**) | 58.7 (**+31.0**) | 14.1 (**+0.6**) | 48.8 (**+9.9**) | 38.6 (**+7.6**) |

Table 7: Best-of-$N$ results on MMLU-Pro using Llama-3.1-8B-Instruct with **label refinement** or **filtering relaxation**. We use R1-Distill-Qwen-14B as the backbone for reward models. The number in parentheses denotes the change after label refinement or filtering relaxation. Please see the caption of Table 5 for more details.

| Method | $N$ | | | | |
|---|---|---|---|---|---|
|  | 1 | 2 | 4 | 8 | 16 |
| Majority voting | 50.27 | 50.27 | 54.15 | 56.14 | 57.16 |
| dORM | 50.27 | 57.30 | 61.54 | 63.95 | 65.38 |
| dPRM | 50.27 | 57.18 | 61.51 | 64.10 | 65.55 |
| dPRM + label refinement | 50.27 | 56.99 (**-0.19**) | 61.41 (**-0.10**) | 64.38 (**+0.28**) | 66.57 (**+1.02**) |
| gORM | 50.27 | 58.24 | 63.88 | 67.82 | 70.02 |
| gPRM | 50.27 | 55.24 | 59.06 | 62.10 | 64.26 |
| gPRM + label refinement | 50.27 | 54.99 (**-0.25**) | 58.86 (**-0.20**) | 62.10 (**+0.00**) | 64.84 (**+0.58**) |
| gPRM + relaxed filtering | 50.27 | 54.93 (**-0.31**) | 58.88 (**-0.18**) | 62.23 (**+0.13**) | 64.82 (**+0.56**) |
| Pass@$N$ | 50.27 | 61.74 | 71.56 | 79.77 | 86.05 |

Table 8: Best-of-$N$ results on MMLU-Pro using CoTs generated by SmolLM3-3B. We use R1-Distill-Qwen-14B backbone for reward models.

| Method | $N$ | | | | |
|---|---|---|---|---|---|
|  | 1 | 2 | 4 | 8 | 16 |
| MV | 40.42 | 40.42 | 44.45 | 46.63 | 48.35 |
| dORM | 40.42 | 48.46 | 54.47 | 58.79 | 61.13 |
| dPRM (min) | 40.42 | 48.00 | 53.80 | 57.95 | 60.32 |
| dPRM (prod) | 40.42 | 47.51 | 53.31 | 57.62 | 60.09 |
| dPRM (mean) | 40.42 | 46.82 | 51.85 | 55.48 | 57.82 |
| dPRM (last) | 40.42 | 48.03 | 53.54 | 57.04 | 58.61 |
| gORM | 40.42 | 49.55 | 56.81 | 62.14 | 65.29 |
| gPRM | 40.42 | 46.34 | 51.72 | 56.29 | 59.66 |
| Pass@$N$ | 40.42 | 53.04 | 65.24 | 76.04 | 83.77 |

Table 9: Best-of-$N$ results on MMLU-Pro using CoTs generated by Qwen2.5-7B-Instruct. We use R1-Distill-Qwen-14B backbone for reward models.

| Method | $N$ | | | | |
| --- | --- | --- | --- | --- | --- |
| | 1 | 2 | 4 | 8 | 16 |
| MV | 56.97 | 56.97 | 59.82 | 61.22 | 62.51 |
| dORM | 56.97 | 62.08 | 65.19 | 66.88 | 67.96 |
| dPRM (min) | 56.97 | 61.89 | 64.98 | 66.83 | 68.02 |
| dPRM (prod) | 56.97 | 61.89 | 64.97 | 66.69 | 67.74 |
| dPRM (mean) | 56.97 | 61.02 | 63.62 | 65.28 | 66.48 |
| dPRM (last) | 56.97 | 61.90 | 64.84 | 66.41 | 67.42 |
| gORM | 56.97 | 62.57 | 66.34 | 68.59 | 70.06 |
| gPRM | 56.97 | 60.88 | 63.80 | 65.75 | 67.44 |
| Pass@$N$ | 56.97 | 66.05 | 73.58 | 79.52 | 84.23 |

Table 10: Best-of-$N$ results on MMLU-Pro using CoTs generated by Llama-3.1-8B-Instruct. We use R1-Distill-Qwen-14B backbone for reward models.

| Method | $N$ | | | | |
| --- | --- | --- | --- | --- | --- |
| | 1 | 2 | 4 | 8 | 16 |
| MV | 50.27 | 50.27 | 54.19 | 56.28 | 57.54 |
| dORM | 50.27 | 57.33 | 61.72 | 63.99 | 64.93 |
| dPRM (min) | 50.27 | 56.93 | 61.19 | 63.63 | 64.88 |
| dPRM (prod) | 50.27 | 56.73 | 60.96 | 63.54 | 64.91 |
| dPRM (mean) | 50.27 | 55.88 | 59.64 | 62.01 | 63.56 |
| dPRM (last) | 50.27 | 57.03 | 61.14 | 63.34 | 64.19 |
| gORM | 50.27 | 58.14 | 63.76 | 67.57 | 69.96 |
| gPRM | 50.27 | 55.13 | 58.93 | 61.96 | 64.34 |
| Pass@$N$ | 50.27 | 61.73 | 71.61 | 79.74 | 85.88 |

Table 11: Best-of-$N$ results on MMLU-Pro using CoTs generated by gemma2-9b-it. We use R1-Distill-Qwen-14B backbone for reward models.

| Method | $N$ | | | | |
| --- | --- | --- | --- | --- | --- |
| | 1 | 2 | 4 | 8 | 16 |
| MV | 52.44 | 52.46 | 54.09 | 54.80 | 55.01 |
| dORM | 52.44 | 56.32 | 58.96 | 60.78 | 61.93 |
| dPRM (min) | 52.44 | 56.17 | 58.66 | 60.18 | 61.27 |
| dPRM (prod) | 52.44 | 56.19 | 58.68 | 60.20 | 61.25 |
| dPRM (mean) | 52.44 | 55.45 | 57.62 | 59.16 | 60.08 |
| dPRM (last) | 52.44 | 56.09 | 58.32 | 59.63 | 60.43 |
| gORM | 52.44 | 56.85 | 60.20 | 62.78 | 64.59 |
| gPRM | 52.44 | 55.74 | 58.46 | 60.64 | 62.22 |
| Pass@$N$ | 52.44 | 59.17 | 64.90 | 69.68 | 73.18 |

Table 12: Best-of-$N$ results on MMLU-Pro using CoTs generated by Llama-3.1-70B-Instruct. We use R1-Distill-Qwen-14B backbone for reward models.

| Method | $N$ | | | | |
|---|---|---|---|---|---|
| | **1** | **2** | **4** | **8** | **16** |
| MV | 65.91 | 65.91 | 69.51 | 71.11 | 72.04 |
| dORM | 65.91 | 70.20 | 72.05 | 72.46 | 72.18 |
| dPRM (min) | 65.91 | 70.06 | 71.92 | 72.60 | 72.63 |
| dPRM (prod) | 65.91 | 69.97 | 71.84 | 72.40 | 72.19 |
| dPRM (mean) | 65.91 | 69.30 | 71.05 | 71.90 | 72.36 |
| dPRM (last) | 65.91 | 70.01 | 71.74 | 72.13 | 71.85 |
| gORM | 65.91 | 71.02 | 73.89 | 75.42 | 76.02 |
| gPRM | 65.91 | 69.22 | 71.38 | 72.88 | 73.79 |
| Pass@$N$ | 65.91 | 74.98 | 81.76 | 86.84 | 90.21 |

