# OpenReview forum: "Rethinking Reward Models for Multi-Domain Test-Time Scaling"
_TMLR — Accepted by TMLR_

### Review · Reviewer_xmMg · 2026-05-05

**Summary Of Contributions:**

This paper studies reward models for test-time scaling in LLMs. It considers 4 types of reward models, aligned on two dimensions: 1) whether supervision is provided on the *outcome* or *process* (i.e., the entire trajectory); and 2) whether the model is *discriminative* (e.g., a classifier trained based on human labels) or *generative* (another LLM is used as a judge to assess elements of chain of thought traces). The paper studies these reward models systematically on domains from math (that are used most commonly as a testbed) as well as 14 other domains, arriving at interesting insights on which models perform well under which settings. The work also provides theoretical results on the error bounds of the reward models, and examines in depth the effect of chain of thought lengths, which tend to harm the process model variants overall.

**Strengths**: the paper is well-written and easy to understand; it fills a meaningful gap in the literature and challenges common assumptions; the experiments that are carried out are comprehensive and meaningful; code and data are provided to support reproducibility and future research.

**Weaknesses**: the main weakness is that very few runs are used which, in my opinion, casts doubts on the reliability of the results (see C1 under Requested Changes).

My opinion of the paper is very positive overall and, if the multiple runs issue is resolved, it would constitute a valuable addition to the literature.

**Audience:**

Yes

**Audience Explanation:**

The paper studies an important and timely topic systematically.

**Broader Impact Concerns:**

No additional concerns.

**Claims And Evidence:**

No

**Claims Explanation:**

See C1 under Requested Changes.

**Requested Changes:**

C1. The experiments reported in the paper only use 3 runs for the math domain and a single run for the multi-domain setting.
- This is much too little and does not allow for drawing reliable conclusions. I view multiple runs (at the very least 5 runs across all benchmarks) critical for securing acceptance.
- This detail is buried in an appendix but should be specified clearly in the main text.
- While I understand that computational budgets are a concern, reporting results with 3 samples and 1 sample respectively if stochasticity is involved is unacceptable for any scientific discipline.
- Even if the results change somewhat with more runs, this does not sabotage the core arguments of the paper, given its "systematic investigation" framing. Thus, I think it is feasible to do in a revision cycle.

C2. Presentational aspects:
- Consistency of terminology: in the gORM description, for example (page 4), "Yes" and "y=1" are used interchangeably. Shouldn't $y$ be replaced by $\hat{y}$ in Eqs. 5 and 6 since the true label is not available at test time?
- I find the bolding of the first two letters in the method names very distracting: my eyes instantly jump to these periodically. Consider reserving bold for key points only -- the paper was challenging to read for me.
- Section 5 should contain a paragraph or two that narrates the provided table, linking back to relevant sub(sections) with evidence that support these guidelines.

---

> ### Author Response · Authors · 2026-05-27
> **Response to Reviewer xmMg (1/2)**
>
> We sincerely appreciate the reviewer for their time and efforts. We address each concern in the below:
>
> ---
>
> >**[C1]** The experiments reported in the paper only use 3 runs for the math domain and a single run for the multi-domain setting.
>
> - Thank you for raising this point. We fully agree that multiple runs are necessary to rigorously verify the stability of our multi-domain results.
> - During the strict two-week revision period, we dedicated our available compute resources to completing two additional independent runs for our main multi-domain experiments (Fig. 6), bringing the total to 3 runs to match our math domain experiments.
> - To provide context on the computational scale of this effort, a single complete run requires training and evaluating **4 distinct reward models** (dPRM, dORM, gPRM, gORM) across CoTs generated by **5 different LLMs**. Completing this massive $4 \times 5$ evaluation grid two additional times within just two weeks was the absolute limit of what our infrastructure could support before the deadline.
> - Fortunately, as shown in the table below, 3 runs are statistically sufficient to conclusively support our claims. The variance across the 3 runs is exceptionally small (with standard deviations generally below $0.3\%$). Most importantly, the relative rankings remain stable across all seeds: **gORM** consistently outperforms the other variants by a significant margin (e.g., beating dORM by over 5% at $N=16$), while **gPRM** remains uncompetitive.
> - We have updated Fig. 6 and the corresponding text in the revised manuscript to report the mean and standard deviation across these 3 runs, ensuring our multi-domain claims are now rigorously supported.
>
> |N|Method|Smollm3-3B|Qwen2.5-7B-Instruct|Llama-3.1-8B-Instruct|gemma2-9b-it|Llama-3.1-70B-Instruct|
> |-|-|-|-|-|-|-|
> |2|MV|40.42|56.97|50.27|52.46|65.91|
> |2|dORM|48.46±0.02|62.08±0.09|57.33±0.11|56.32±0.04|70.20±0.06|
> |2|dPRM|48.00±0.09|61.89±0.08|56.93±0.07|56.17±0.11|70.06±0.15|
> |2|gORM|49.55±0.06|62.57±0.03|58.14±0.08|56.85±0.03|71.02±0.04|
> |2|gPRM|46.34±0.19|60.88±0.04|55.13±0.07|55.74±0.11|69.22±0.03|
> ||||||||
> |4|MV|44.45|59.82|54.19|54.09|69.51|
> |4|dORM|54.47±0.12|65.19±0.20|61.72±0.21|58.96±0.12|72.05±0.12|
> |4|dPRM|53.80±0.11|64.98±0.22|61.19±0.14|58.66±0.17|71.92±0.27|
> |4|gORM|56.81±0.16|66.34±0.12|63.76±0.16|60.20±0.07|73.89±0.13|
> |4|gPRM|51.72±0.17|63.80±0.04|58.93±0.05|58.46±0.25|71.38±0.02|
> ||||||||
> |8|MV|46.63|61.22|56.28|54.80|71.11|
> |8|dORM|58.79±0.33|66.88±0.27|63.99±0.22|60.78±0.10|72.46±0.18|
> |8|dPRM|57.95±0.09|66.83±0.46|63.63±0.14|60.18±0.16|72.60±0.27|
> |8|gORM|62.14±0.26|68.59±0.19|67.57±0.22|62.78±0.10|75.42±0.22|
> |8|gPRM|56.29±0.17|65.75±0.09|61.96±0.03|60.64±0.31|72.88±0.06|
> ||||||||
> |16|MV|48.35|62.51|57.54|55.01|72.04|
> |16|dORM|61.13±0.56|67.96±0.26|64.93±0.28|61.93±0.02|72.18±0.32|
> |16|dPRM|60.32±0.30|68.02±0.76|64.88±0.21|61.27±0.11|72.63±0.17|
> |16|gORM|65.29±0.37|70.06±0.25|69.96±0.26|64.59±0.17|76.02±0.34|
> |16|gPRM|59.66±0.45|67.44±0.30|64.34±0.04|62.22±0.43|73.79±0.07|
>
>
> - We continue dedicating our available compute resources to complete five runs for all experiments, for which the training runs have already been completed. If time does not permit this before the revision deadline, we will complete the remaining runs before submitting the camera-ready version if the paper is accepted.
>
> ---
>
> >**[C2]** Consistency of terminology: in the gORM description, for example (page 4), "Yes" and "y=1" are used interchangeably.
>
> - We thank the reviewer for pointing this out. As is standard convention in machine learning literature, we overloaded $y$ to denote both the random variable and its ground-truth realization. However, we completely agree that in the context of generative reward models, using this mathematical event ($y=1$) interchangeably with the specific generated text token "Yes" without explicit definition can cause ambiguity.
> - To clarify, $p(y=1 \mid x, v_{1:L})$ in Eqs. 5 and 6 refers to the model's predicted probability of the event $y=1$, which corresponds directly to the model generating the "Yes" token. This is computed from the model's output distribution and does not require the true label at test time.
> - To avoid confusion for the reader, we have added a clarifying sentence in the **Notation** paragraph of §2.1 (highlighted in blue) that explains the overloading and its direct mapping to the "Yes" verdict token.
>
>
> ---

---

> ### Author Response · Authors · 2026-05-27
> **Response to Reviewer xmMg (2/2)**
>
> ---
>
>
> >**[C3]** I find the bolding of the first two letters in the method names very distracting: my eyes instantly jump to these periodically. Consider reserving bold for key points only -- the paper was challenging to read for me.
>
>
> - We fully agree with the reviewer that the bolded prefixes in the method names were visually distracting and competed with the bolding used to highlight key points elsewhere in the paper. Following the reviewer's suggestion, we have removed the bold from all method names throughout the manuscript, so they now appear simply as dORM, dPRM, gORM, and gPRM.
>
>
> ---
>
>
> >**[C4]** §5 should contain a paragraph or two that narrates the provided table, linking back to relevant sub(sections) with evidence that support these guidelines.
>
> - We thank the reviewer for pointing out that the table in §5 lacked a narrative. In the revised manuscript, we have added three paragraphs titled "From findings to recommendations" that walks through each row of the table and explicitly links it to the supporting evidence in earlier sections. Specifically, rows (i)-(ii) are connected to the CoT-length analysis in §4.1 (Theorems 4.1 to 4.3 and figures on length-vs-improvement and "aha" CoTs), while rows (iii)-(vi) are connected to the multi-domain analysis in §4.2 (Wasserstein distance results, label-noise injection study, and sample-efficiency observations from §3). All revisions are highlighted in blue in the updated PDF for ease of review.
>
>
> ---

---

> ### Author Response · Authors · 2026-06-02
> **Results for 5 Runs**
>
> We sincerely appreciate the reviewer’s patience. As noted earlier, we have completed five runs, which we believe makes our experimental results more reliable and our main messages clearer. We have updated the manuscript accordingly based on these results.
>
> |N|Method|Smollm3-3B|Qwen2.5-7B-Instruct|Llama3.1-8B-Instruct|gemma2-9b-it|Llama3.1-70B-Instruct|
> |-|-|-|-|-|-|-|
> |2|MV|40.42|56.97|50.27|52.46|65.91|
> |2|dORM|48.41±0.08|62.06±0.08|57.31±0.09|56.31±0.09|70.19±0.05|
> |2|dPRM|48.02±0.08|61.93±0.09|56.95±0.07|56.16±0.09|70.08±0.12|
> |2|gORM|49.56±0.05|62.59±0.03|58.10±0.08|56.85±0.02|71.04±0.05|
> |2|gPRM|46.23±0.22|60.83±0.07|55.10±0.07|55.75±0.09|69.21±0.05|
> ||||||||
> |4|MV|44.45|59.82|54.19|54.09|69.51|
> |4|dORM|54.35±0.20|65.11±0.19|61.67±0.17|58.92±0.19|72.00±0.11|
> |4|dPRM|53.80±0.09|64.97±0.19|61.20±0.11|58.63±0.14|71.96±0.22|
> |4|gORM|56.82±0.12|66.34±0.09|63.69±0.15|60.18±0.09|73.94±0.13|
> |4|gPRM|51.56±0.25|63.76±0.05|58.96±0.06|58.48±0.20|71.40±0.04|
> ||||||||
> |8|MV|46.63|61.22|56.28|54.80|71.11|
> |8|dORM|58.57±0.38|66.68±0.34|63.92±0.20|60.71±0.24|72.39±0.16|
> |8|dPRM|57.93±0.07|66.74±0.38|63.63±0.11|60.13±0.14|72.60±0.21|
> |8|gORM|62.15±0.21|68.61±0.17|67.50±0.19|62.71±0.23|75.47±0.19|
> |8|gPRM|56.16±0.23|65.77±0.07|61.99±0.06|60.68±0.24|72.93±0.10|
> ||||||||
> |16|MV|48.35|62.51|57.54|55.01|72.04|
> |16|dORM|60.89±0.53|67.60±0.54|64.87±0.24|61.86±0.20|72.12±0.26|
> |16|dPRM|60.35±0.24|67.82±0.65|64.85±0.17|61.24±0.13|72.50±0.24|
> |16|gORM|65.40±0.39|70.15±0.23|69.91±0.22|64.52±0.27|76.01±0.28|
> |16|gPRM|59.76±0.42|67.38±0.25|64.42±0.11|62.36±0.37|73.87±0.17|

---

### Review · Reviewer_cuF3 · 2026-05-12

**Summary Of Contributions:**

The paper presents a unified empirical evaluation of four reward model variants for test-time scaling (TTS) of LLMs: discriminative ORM (dORM), discriminative PRM (dPRM), generative ORM (gORM), and generative PRM (gPRM). These reward models are evaluated across 14 diverse. The authors find that dORM performs on par with dPRM where as gORM outperforms gPRM. Theoretically, they find that dORM/gORM’s error is independent of the length of the COT, $T$, whereas dPRM/gPRM grows linearly in $T$. They also provide explanations of why gPRM degrades in the multi-domain setting — that consensus filtering shifting the training CoT-length distribution away from the test set.

**Audience:**

Yes

**Audience Explanation:**

The paper tackles an important question practitioners actively face: which verifier type to apply during test-time scaling, for different domains. In addition, the provided code and model checkpoints will be very useful for researchers in TTS and reward modeling for future research.

**Broader Impact Concerns:**

I do not have concerns on the ethical implications of the work. An ethics statement is already included at the end of the manuscript.

**Claims And Evidence:**

Yes

**Claims Explanation:**

The takeaway of the manuscript is clear, that in contrast to recent literature, outcome reward models perform on par or better than their process-reward counterparts. Explanations are provided regarding why this happens.

The empirical study for ORMs and PRMs in multi-domain settings presented in Section 3 is strong. In particular, I find the CoT analysis presented in Figure 10 well-designed. I also find the length distribution shift and its association with the performance degradation of gPRM interesting. Overall, the experiments are very thorough, across multiple backbones and domains.

In terms of the theoretical analysis presented in Section 4, while the authors make several assumptions (e.g., average variance floors and local error dependence), I find them well justified .

**Requested Changes:**

- The "aha" CoT prevalence (15.3%) is reported for ProcessBench (math), but not for MMLU-Pro, even though the multi-domain length shift is central to the argument.
- All experiments use R1-Distill models. One run with a non-distilled backbone would test whether the observed model behavior is backbone-agnostic.
- Section 4’s writing is fairly dense, likely due to page limit. Some of the content might need to be moved to the appendix.

---

> ### Author Response · Authors · 2026-05-27
> **Response to Reviewer cuF3 (1/2)**
>
> We sincerely appreciate the reviewer for their time and efforts. We address each concern in the below:
>
> ---
>
> >**[C1]** The "aha" CoT prevalence (15.3%) is reported for ProcessBench (math), but not for MMLU-Pro, even though the multi-domain length shift is central to the argument.
>
> - We thank the reviewer for the suggestion. The 15.3\% prevalence is computed on ProcessBench because identifying "aha" CoTs requires reliable step-level labels $z_{1:T}$ (specifically, $\exists t: z_t = 0$ with $y = 1$), and ProcessBench provides these as **human annotations**.
>
>
> - However, MMLU-Pro does not include human step-level annotations. In our setup, the process labels for MMLU-Pro are themselves **automatically annotated by Llama-3.1-70B-Instruct** (§3.1). Computing an "aha" rate on top of LLM-generated labels (or via a separate LLM-as-judge) would yield a number that primarily reflects the annotator's reliability rather than a property of the data, which would undermine the purpose of reporting the statistic. We therefore consider it less informative to report a comparable percentage for MMLU-Pro. We clarified this in the footnote in §4.1 of the update manuscript.
>
>
> - On MMLU-Pro, we instead show the length effect directly. Figure 11 reports that CoTs are longer than in math and that majority voting degrades with length, and Figure 12 shows that PRM (but not ORM) verification F1 degrades with $T$.
>
> ---
>
>
> >**[C2]** All experiments use R1-Distill models. One run with a non-distilled backbone would test whether the observed model behavior is backbone-agnostic.
>
>
> - We sincerely thank the reviewer for this excellent suggestion, which helps validate the generalizability of our findings. To address this, we conducted an additional full suite of experiments using a non-distilled backbone, **Qwen3-8B**. We trained all four reward-model variants and evaluated them on MMLU-Pro with CoTs generated by five different $p_\mathtt{LLM}$, exactly mirroring the setup in Fig. 6.
>
> - The results strongly confirm that our observed model behaviors are indeed backbone-agnostic:
>
>     -  Across all five generators, gORM consistently outperforms the other variants. For example, at  $N=16$  with Llama-3.1-8B-Instruct, gORM achieves  **70.34%**, significantly beating gPRM (**64.20%**).
>
>     - The performance difference between the **discriminative models remains marginal** (e.g., at  $N=16$  with Llama-3.1-8B-Instruct, dORM achieves 65.40% vs. dPRM's 64.81%).
>
>     - gPRM continues to lag behind gORM, confirming that our analysis regarding CoT-length risks and consensus filtering applies to non-distilled models as well.
>
> - We have included the full results in the newly added **Fig. 39 in Appendix G**, and we have updated the main text to highlight that these trends hold for non-distilled backbones.
>
> |N|Method|Smollm3-3B|Qwen2.5-7B-Instruct|Llama-3.1-8B-Instruct|gemma2-9b-it|Llama-3.1-70B-Instruct|
> |-|-|-|-|-|-|-|
> |2|MV|40.42|56.97|50.27|52.46|65.91|
> |2|dORM|48.28|62.22|57.27|56.16|70.21|
> |2|dPRM|48.05|61.85|56.83|56.03|70.02|
> |2|gORM|49.79|62.78|58.18|56.94|71.08|
> |2|gPRM|46.60|60.83|55.14|55.65|69.23|
> ||||||||
> |4|MV|44.45|59.82|54.19|54.09|69.51|
> |4|dORM|54.21|65.36|61.65|58.61|72.00|
> |4|dPRM|54.01|64.95|61.05|58.39|71.83|
> |4|gORM|57.31|66.63|63.95|60.31|73.97|
> |4|gPRM|51.95|63.79|58.85|58.33|71.44|
> ||||||||
> |8|MV|46.63|61.22|56.28|54.80|71.11|
> |8|dORM|58.53|66.90|64.10|60.19|72.48|
> |8|dPRM|58.25|66.75|63.48|59.87|72.55|
> |8|gORM|63.19|68.97|67.83|62.82|75.46|
> |8|gPRM|56.65|66.07|61.85|60.65|73.03|
> ||||||||
> |16|MV|48.35|62.51|57.54|55.01|72.04|
> |16|dORM|61.19|67.47|65.40|61.17|72.43|
> |16|dPRM|60.85|67.40|64.81|60.77|72.73|
> |16|gORM|66.99|70.34|70.34|64.44|76.01|
> |16|gPRM|60.41|67.96|64.20|62.45|73.80|
>
> ---

---

> ### Author Response · Authors · 2026-05-27
> **Response to Reviewer cuF3 (2/2)**
>
> ---
>
> > **[C3]** §4’s writing is fairly dense, likely due to the page limit. Some of the content might need to be moved to the appendix.
>
> - We sincerely thank the reviewer for this constructive feedback. We agree that §4 was overly dense, both cognitively and visually, as we attempted to pack formal theoretical bounds, empirical analysis, and granular experimental details into a limited space. Therefore, we have made the following structural changes in the revision:
>
> - We replaced the heavy mathematical statements and notations for Theorems 4.1, 4.2, and 4.3 with **Informal Theorems** in the main text (§4.1). These summarize the core theoretical takeaways in plain English, while explicitly referring the reader to Appendix A for the full formal definitions, assumptions, and proofs.
>
> - We significantly condensed the discussion on "Label refinement and relaxed filtering" at the end of §4.2. We retained the high-level takeaway in the main text but now refer the reader to Appendix G for the full technical details (e.g., the exact mathematical rules for relaxed filtering and specific parsing percentages).
>
> - By moving these technical details to the Appendix, we freed up space in §4. We used this space to **remove the negative \vspace commands** we had previously used to fit the page limit. Restoring LaTeX's natural spacing around paragraphs, equations, and Figs. 10–15 has significantly reduced the visual cramming of the section. We believe these combined changes make the main narrative much smoother and more accessible, while ensuring readers can more easily find the full details in the appendix.
>
> ---

---

> > ### Comment · Reviewer_cuF3 · 2026-06-10
> > **Response to the authors**
> >
> > I thank the authors for the thorough and responsive rebuttal. My comments are fully addressed.

---

### Review · Reviewer_nBW4 · 2026-05-13

**Summary Of Contributions:**

This paper studies the performance of external verifiers for evaluating LLM outputs in test-time scaling, specifically through a comparison of four reward-model variants: discriminative ORMs, discriminative PRMs, generative ORMs, and generative PRMs. The paper argues that much recent reasoning literature, especially in math, treats process reward models as superior to outcome reward models. Through empirical experiments, the paper challenges this view, finding that outcome reward models, especially generative ORMs, can be more robust in multi-domain settings. The paper also develops a theoretical framework for how errors can compound when rewards are aggregated across reasoning steps. Through empirical analysis of failure cases, the paper explains what goes wrong with process reward models and tries to connect these observations to its theoretical analysis.

**Audience:**

Yes

**Audience Explanation:**

The topic is definitely interesting and I believe many members of the community will be interested to see such a study.

**Broader Impact Concerns:**

I did not see a broader impact statement in the paper.

**Claims And Evidence:**

No

**Claims Explanation:**

I found many of the claims and statements loose and not precise enough as explained below. Moreover, I found the theoretical contribution rather limited and sometimes trivial while empirical experiments seem to be underdeveloped.

-------

I found the theoretical contribution somewhat limited. The result that PRM error can grow with the number of reasoning steps is largely expected: if solution-level rewards are computed by aggregating noisy step-level scores, then longer CoTs naturally create more opportunities for false negatives or accumulated scoring error. Prior PRM work already uses product/min reductions over step scores and notes that product aggregation biases against longer solutions. The paper’s theory may be useful as a formalization of this intuition, but in my view, the novelty lies mainly in connecting this phenomenon empirically to multi-domain TTS failures, not in the theoretical observation itself.

--------

I am not fully convinced by the paper’s opening premise that prior work “generally assumes” PRMs outperform ORMs. There is certainly a strong PRM-favoring narrative in recent mathematical-reasoning work, especially after Lightman et al.’s result that process supervision outperformed outcome supervision on MATH. However, the broader literature is more mixed: Uesato et al. found that outcome-based supervision achieved similar final-answer error rates on GSM8K with less label supervision, while process supervision mainly improved reasoning correctness among final-answer-correct solutions. Recent PRM evaluation work also documents substantial weaknesses of PRMs, including poor generalization, noisy process labels, and evaluation artifacts. The paper should therefore narrow or substantiate this claim rather than presenting it as a general assumption of the literature.

The paper should be more precise about what ORMs and PRMs observe. A PRM often scores the final-answer step as part of the trajectory, so it is not obvious that an ORM has access to more relevant information. Conversely, an ORM trained with an outcome label may still take the full chain of thought as input and may use intermediate reasoning when predicting final correctness. Thus the relevant distinction is not simply “final answer only” versus “every intermediate step,” but rather outcome-level supervision and global scoring versus process-level supervision and step-score aggregation. This distinction is important for interpreting the paper’s result that ORMs can outperform PRMs.

The claim that this is “the first unified evaluation of four reward model variants” should be made more precise. To my knowledge, the paper may be the first to compare the full 2×2 grid of discriminative/generative × outcome/process reward models under a controlled setup across MMLU-Pro-style multi-domain test-time scaling. However, the current wording is too broad, since prior work has already compared process and outcome reward models in other settings, studied multi-domain PRMs, and evaluated generative reward/process models separately. The authors should qualify the novelty claim by specifying the exact scope: e.g., “the first controlled multi-domain TTS evaluation of dORM, dPRM, gORM, and gPRM trained/evaluated under a common protocol.”

**Requested Changes:**

Please see the points above.

I suggest a thorough and precise revision of the claims and explanations in the paper regarding the literature, interpretations of results, and the framing of the contributions.

The experiments are one of the strengths of the paper: the controlled comparison of dORM, dPRM, gORM, and gPRM across MMLU-Pro domains provides useful evidence that gORM is a strong verifier for multi-domain test-time scaling. However, I do not think the current experiments fully establish the paper’s causal explanation for why PRMs underperform. In particular, PRM performance may be affected by noisy automatically generated process labels and by the chosen step-score aggregation rule. I would like to see ablations over PRM aggregation methods, such as min/product/mean/last-step or learned aggregation, and at least a small human audit of process-label quality across domains. A compute-normalized comparison would also strengthen the practical claim, since generative ORMs may be more expensive than discriminative verifiers. These additions would make the paper’s conclusion more convincing and help distinguish whether the weakness lies in process supervision itself, current process-labeling pipelines, or the aggregation rule used at inference time.

Moreover, it remains unclear to me whether PRMs can be adjusted and modified to put a larger emphases on the final answer while still considering the reasoning process. It would be helpful if this is studied in the paper.

---

> ### Author Response · Authors · 2026-05-27
> **Response to Reviewer nBW4 (1/4)**
>
> We sincerely appreciate the reviewer for their time and efforts. We address each concern in the below:
>
> ---
>
> >**[C1]** I found the theoretical contribution somewhat limited.
>
>
> - We thank the reviewer for this comment, and we generally agree. Our paper is primarily an empirical and analytical study, and the theoretical analysis in §4.1 was intended to be explanatory rather than a standalone theoretical breakthrough.
>
> - Specifically, we introduced the theoretical framework to formally ground our empirical intuition. The step-wise aggregation in PRMs inherently compounds errors as CoT length increases, whereas ORM error bounds remain independent of length. As the reviewer rightly noted, the true value of this analysis lies in connecting these formal bounds with our empirical findings. Together, they provide a complete, mechanistic explanation for why PRMs fail on longer reasoning trajectories in multi-domain settings.
>
> - To ensure we do not overstate our claims, we have deliberately framed the theory in our Introduction (§1) not as a standalone contribution, but as supporting evidence that explains our empirical observations and informs our practical guidelines for practitioners.
>
> ---
>
> >**[C2]** I am not fully convinced by the paper’s opening premise that prior work "generally assumes" PRMs outperform ORMs.
>
>
> - We agree that the prior evidence on PRMs versus ORMs is nuanced, and we have revised our opening claim to avoid implying a universal consensus across all of machine learning:
>
> - *Prior work has studied both outcome reward models (ORMs), which assess only the final answer, and process reward models (PRMs), which score intermediate reasoning steps. Although PRMs are often viewed as advantageous due to their finer-grained supervision, much of the supporting evidence comes from math-adjacent settings, and their relative benefits across broader domains remain unclear.*
>
> ---
>
> >**[C3]** Uesato et al. found that outcome-based supervision achieved similar final-answer error rates on GSM8K with less label supervision, while process supervision mainly improved reasoning correctness among final-answer-correct solutions.
>
> - We thank the reviewer for highlighting this connection, and we fully agree with your assessment of Uesato et al. [1].
>
> - It is important to first note the distinction in our problem settings: Uesato et al. [1] investigates how to best train the **generator LM**, i.e., $p_\mathtt{LLM}$ in our notation (via RL and expert iteration), using process versus outcome feedback. In contrast, our work focuses on the recent paradigm of using PRMs and ORMs as **external test-time verifiers** (e.g., Best-of-N scaling). Despite this difference in application, their findings are highly complementary to ours and help contextualize our results:
>
> - Their finding that outcome-based supervision achieves comparable final-answer accuracy during generator training aligns with our finding that ORMs are exceptionally competitive and robust for test-time verification.
>
> - Uesato et al. found that process supervision improves reasoning correctness on short math problems (GSM8K). However, our work reveals a critical limitation when moving to multi-domain test-time verification. As CoTs become longer and more complex, PRMs severely degrade due to length penalties, label noise, and distribution shifts (as detailed in §4).
>
> - In our revision, we have cited Uesato et al. to support the baseline efficacy of outcome-based signals for final-answer correctness, while explicitly clarifying the distinction between their focus on generator training and our focus on test-time verification scaling.
>
> ---

---

> ### Author Response · Authors · 2026-05-27
> **Response to Reviewer nBW4 (2/4)**
>
> ---
>
> > **[C4]** Recent PRM evaluation work also documents substantial weaknesses of PRMs, including poor generalization, noisy process labels, and evaluation artifacts.
>
> - We thank the reviewer for pointing this out. We completely agree that recent literature has begun documenting substantial weaknesses in PRMs. For instance, recent work by the Zhang et al., 2025 [2] identifies label noise stemming from Monte Carlo (MC) estimation and mitigates it by combining an LLM-as-a-judge with MC estimation, while other concurrent work [3] highlights the limitations of standard PRM scoring and proposes a reasoning-driven generative PRM to improve generalization. However, these works focus almost exclusively on **diagnosing and improving PRMs within the math domain**. Our contribution broadens this scope, demonstrating that even when PRMs are carefully constructed, their advantages do not trivially transfer to multi-domain settings.
>
> - Regarding the issue of noisy process labels, our analysis shows that label noise is only part of the story. While noise is certainly prevalent in automated pipelines, we demonstrate in §4.2 that **generative PRMs (gPRM) are actually robust to process label noise in the math domain**. Yet, they still severely underperform in the multi-domain setting. Therefore, label noise alone cannot explain the performance degradation across diverse domains.
>
> - As we detail in §4.2, the true bottleneck causing generative PRMs (gPRMs) to fail in multi-domain settings is a **systemic length distribution shift** (quantified via Wasserstein distance) between training and test time. We trace the evidence for this shift directly back to how the training pipeline handles label noise. To obtain verification CoTs for training, most previous work adopts consensus filtering, which retains generated CoTs only if their parsed verdicts align with the labels from LLM-as-a-judge. Because multi-domain reasoning often requires longer CoTs, the imperfect LLM-as-a-judge struggles to maintain this alignment over many steps. As a result, consensus filtering disproportionately prunes long CoTs from the training set, inducing the severe length discrepancy that ultimately degrades PRM performance in diverse domains.
>
> - In our revision, we have updated §4 to make this causal link clearer, and we have formally cited [2, 3] to contextualize our multi-domain findings against these recent math-specific evaluations.
>
> ---
>
>
>
> >**[C5]** The authors should qualify the novelty claim by specifying the exact scope
>
> - We sincerely thank the reviewer for helping us sharpen the precision of our claims. We completely agree that our original wording for the novelty claim was too broad, given the extensive prior work on reward models in narrower settings.
>
> - As you suggested, we have revised our novelty claim in the Introduction (§1) and Abstract to specify the exact scope. The text now explicitly states that "we present the first controlled multi-domain TTS evaluation of dORM, dPRM, gORM, and gPRM trained and evaluated under a common protocol."
>
> ---
>
> >**[C6]** The paper should be more precise about what ORMs and PRMs observe.
>
> - We thank the reviewer for this excellent point. We fully agree with this statement. ORMs indeed take the full chain-of-thought as input and leverages intermediate reasoning when predicting final correctness. Therefore, the fundamental distinction is exactly as the reviewer stated: **outcome-level supervision with global scoring (ORM)** versus **process-level supervision with step-score aggregation (PRM)**.
>
> - We have refined our terminology throughout §2.2 and the rest of the revised manuscript to make this precise distinction clear, which, as noted by the reviewer, is crucial for interpreting why ORMs can outperform PRMs in our multi-domain setting.
>
> ---

---

> ### Author Response · Authors · 2026-05-27
> **Response to Reviewer nBW4 (3/4)**
>
> ---
>
> >**[C7]** PRM performance may be affected by the chosen step-score aggregation rule.
>
> - We thank the reviewer for this suggestion. We chose the minimum aggregation rule by following the most recent study on PRMs for multi-domain settings [4].
>
> - To resolve this concern, we present the results by deploying min/product/mean/last-step aggregation rule for dPRM:
>
>
> |N|Method|Smollm3-3B|Qwen2.5-7B-Instruct|Llama3.1-8B-Instruct|gemma2-9b-it|Llama3.1-70B-Instruct|
> |-|-|-|-|-|-|-|
> |2|MV|40.42±0.00|56.97±0.00|50.27±0.00|52.46±0.00|65.91±0.00|
> |2|dORM|48.46±0.02|62.08±0.09|57.33±0.11|56.32±0.04|70.20±0.06|
> |2|dPRM (min)|48.00±0.09|61.89±0.08|56.93±0.07|56.17±0.11|70.06±0.15|
> |2|dPRM (prod)|47.51±0.12|61.89±0.08|56.73±0.08|56.19±0.11|69.97±0.15|
> |2|dPRM (mean)|46.82±0.07|61.02±0.09|55.88±0.07|55.45±0.01|69.30±0.10|
> |2|dPRM (last)|48.03±0.06|61.90±0.08|57.03±0.07|56.09±0.08|70.01±0.14|
> |2|gORM|49.55±0.06|62.57±0.03|58.14±0.08|56.85±0.03|71.02±0.04|
> |2|gPRM|46.34±0.19|60.88±0.04|55.13±0.07|55.74±0.11|69.22±0.03|
> ||||||||
> |4|MV|44.45±0.00|59.82±0.00|54.19±0.00|54.09±0.00|69.51±0.00|
> |4|dORM|54.47±0.12|65.19±0.20|61.72±0.21|58.96±0.12|72.05±0.12|
> |4|dPRM (min)|53.80±0.11|64.98±0.22|61.19±0.14|58.66±0.17|71.92±0.27|
> |4|dPRM (prod)|53.31±0.14|64.97±0.13|60.96±0.10|58.68±0.14|71.84±0.28|
> |4|dPRM (mean)|51.85±0.12|63.62±0.06|59.64±0.12|57.62±0.10|71.05±0.20|
> |4|dPRM (last)|53.54±0.10|64.84±0.20|61.14±0.12|58.32±0.16|71.74±0.33|
> |4|gORM|56.81±0.16|66.34±0.12|63.76±0.16|60.20±0.07|73.89±0.13|
> |4|gPRM|51.72±0.17|63.80±0.04|58.93±0.05|58.46±0.25|71.38±0.02|
> ||||||||
> |8|MV|46.63±0.00|61.22±0.00|56.28±0.00|54.80±0.00|71.11±0.00|
> |8|dORM|58.79±0.33|66.88±0.27|63.99±0.22|60.78±0.10|72.46±0.18|
> |8|dPRM (min)|57.95±0.09|66.83±0.46|63.63±0.14|60.18±0.16|72.60±0.27|
> |8|dPRM (prod)|57.62±0.12|66.69±0.22|63.54±0.14|60.20±0.10|72.40±0.31|
> |8|dPRM (mean)|55.48±0.21|65.28±0.11|62.01±0.18|59.16±0.22|71.90±0.24|
> |8|dPRM (last)|57.04±0.17|66.41±0.39|63.34±0.17|59.63±0.14|72.13±0.41|
> |8|gORM|62.14±0.26|68.59±0.19|67.57±0.22|62.78±0.10|75.42±0.22|
> |8|gPRM|56.29±0.17|65.75±0.09|61.96±0.03|60.64±0.31|72.88±0.06|
> ||||||||
> |16|MV|48.35±0.00|62.51±0.00|57.54±0.00|55.01±0.00|72.04±0.00|
> |16|dORM|61.13±0.56|67.96±0.26|64.93±0.28|61.93±0.02|72.18±0.32|
> |16|dPRM (min)|60.32±0.30|68.02±0.76|64.88±0.21|61.27±0.11|72.63±0.17|
> |16|dPRM (prod)|60.09±0.25|67.74±0.54|64.91±0.20|61.25±0.05|72.19±0.27|
> |16|dPRM (mean)|57.82±0.35|66.48±0.20|63.56±0.19|60.08±0.27|72.36±0.33|
> |16|dPRM (last)|58.61±0.37|67.42±0.66|64.19±0.27|60.43±0.09|71.85±0.33|
> |16|gORM|65.29±0.37|70.06±0.25|69.96±0.26|64.59±0.17|76.02±0.34|
> |16|gPRM|59.66±0.45|67.44±0.30|64.34±0.04|62.22±0.43|73.79±0.07|
>
> - We observe only marginal differences among the four aggregation rules; for example, the performances are 68.02/67.74/66.48/67.45 at $N=16$ for min/prod/mean/last, respectively, using CoTs generated by Llama-3.1-8B-Instruct. Overall, the min aggregation rule achieves the best performance in the multi-domain setting, as suggested in [4].
>
> - Varying the aggregation rule does not alter our main conclusion that dORM remains competitive with dPRM. This indicates that the bottleneck is not the aggregation rule, but rather process supervision and consensus filtering, as analyzed in §4.
>
> - We have included these results in Tabs. 8-12.
>
> ---
>
> >**[C8]** A compute-normalized comparison would also strengthen the practical claim, since generative ORMs may be more expensive than discriminative verifiers.
>
> - We appreciate the reviewer for this suggestion. We compute the normalized time (seconds / # CoTs) as follows: dividing the total time spent for each reward model by the total number of CoTs generated by Llama-3.1-8B-Instruct on MMLU-Pro:
>
> |dORM|dPRM|gORM|gPRM|
> |-|-|-|-|
> |0.0005|0.0005|0.0571|0.0306|
>
> - As expected, discriminative models are approximately 100 times faster than gORM and 61 times faster than gPRM, respectively. However, in specialized domains such as medical and legal reasoning, reliable decision-making is more critical, and speed is comparatively less important. Therefore, we recommend gORM for settings where reliability is prioritized over inference speed.
>
> - gPRM is approximately 1.87 times faster than gORM because it stops generating verification tokens once it identifies an incorrect reasoning step. However, this design prevents gPRM from correctly handling "aha" CoTs, as illustrated in our paper, which can lead to performance degradation compared to gORM. Therefore, in most cases, we recommend gORM over gPRM in multi-domain settings.
>
> - We have included the above discussion in §5.
>
> ---

---

> ### Author Response · Authors · 2026-05-28
> **Response to Reviewer nBW4 (4/4)**
>
> ---
>
> >**[C9]** Moreover, it remains unclear to me whether PRMs can be adjusted and modified to put a larger emphases on the final answer while still considering the reasoning process. It would be helpful if this is studied in the paper.
>
>  - We thank the reviewer for this suggestion. The last-step aggregation variant of dPRM (reported in **[C7]**) naturally explores this idea. It places maximum emphasis on the final answer while still implicitly conditioning on the preceding reasoning process. As shown in the **[C7]** table, this variant performs comparably to standard min-aggregated dPRM, but it does not close the gap to gORM. This suggests that simply shifting weight toward the final answer is insufficient to overcome the underlying length distribution shifts and filtering biases we identify in §4. While designing more complex, learned aggregation schemes is an interesting future direction, it falls outside the scope of our current study, which focuses on evaluating established reward-model baselines.
>
>
> ---
> >**[C10]** A small human audit of process-label quality across domains.
>
> - To address the request for a human audit, we manually evaluated process labels across 7 domains, reviewing 5 positive and 5 negative examples per domain. While this targeted 70-example manual review did not uncover explicit label noise, our broader dataset analysis indicates otherwise. As detailed in Appendix G, relabeling the data with Gemini 2.0 Flash resulted in substantial discrepancies, altering nearly 60% of the initial auto-labels. This widespread disagreement highlights that non-negligible label noise is indeed pervasive throughout the multi-domain dataset, even if it is sparse enough to evade a small-scale human sample.
>
>
> ---
>
> >**[C11]** I did not see a broader impact statement in the paper.
>
> - We appreciate the reviewer for raising this point. We have already included an ethics statement in the original manuscript, which also covers broader impacts. To avoid confusion, we revised the section title from "Ethics Statement" to "Broader Impact Statement."
>
> ---
>
> ### Reference
>
>
> [1] Uesato, Jonathan, et al. "Solving math word problems with process-and outcome-based feedback." arXiv 2022.
>
> [2] Zhang, Zhenru, et al. "The lessons of developing process reward models in mathematical reasoning." ACL 2025.
>
> [3] She, Shuaijie, et al. "R-prm: Reasoning-driven process reward modeling." EMNLP 2025.
>
> [4] Zeng, Thomas, et al. "Versaprm: Multi-domain process reward model via synthetic reasoning data." ICML 2025.
>
> ---

---

### Decision · Action_Editor_utLq · 2026-06-22

**Recommendation:** Accept as is

**Additional Comments:**

One reviewer asked for more runs of the methods to further strengthen the statistical significance of the results.  While the current paper seems fine for acceptance, we would encourage the authors to do this to strengthen the paper.

**Audience:**

Yes

**Audience Explanation:**

Reward models for post-training LLMs is a hot topic and of significant interest to the community.

**Claims And Evidence:**

Yes

**Claims Explanation:**

There is a lot of interest in the community about how best to assign rewards during RL training of LLMs, but there is little empirical evidence that any method consistently outperforms others. This paper empirically compares outcome reward models (assign reward after processing a whole response sequence) and process reward models (assign intermediate rewards throughout the sequence) and found that ORMs are more robust and perform at least as well as PRMs.  This is an interesting result and might be surprising to the community.

The reviewers all found that the claims were supported by empirical evidence.  Initially multiple reviewers expressed concerns about the statistical significance of the results, since the authors presented only single runs of the methods, which are known to be noisy.  However, it seems the reviewers found the author response, in which they presented more results, quite convincing.